# Mechanisms and Regulation of Cellular Senescence

**DOI:** 10.3390/ijms222313173

**Published:** 2021-12-06

**Authors:** Lauréline Roger, Fanny Tomas, Véronique Gire

**Affiliations:** 1Structure and Instability of Genomes Laboratory, Muséum National d’Histoire Naturelle (MNHN), CNRS-UMR 7196/INSERM U1154, 43 Rue Cuvier, 75005 Paris, France; laureline.roger@mnhn.fr; 2Centre de Recherche en Biologie cellulaire de Montpellier (CRBM), Université de Montpellier, CNRS UMR 5237, 1919 Route de Mende, 34293 Montpellier, France; TomasF@mskcc.org

**Keywords:** cellular senescence, cell cycle arrest, DNA damage signaling, transcriptome signature, senescence-associated secretory phenotype, mitochondrial dysfunction, metabolism alteration, epigenetic and chromatin changes, aging

## Abstract

Cellular senescence entails a state of an essentially irreversible proliferative arrest in which cells remain metabolically active and secrete a range of pro-inflammatory and proteolytic factors as part of the senescence-associated secretory phenotype. There are different types of senescent cells, and senescence can be induced in response to many DNA damage signals. Senescent cells accumulate in different tissues and organs where they have distinct physiological and pathological functions. Despite this diversity, all senescent cells must be able to survive in a nondividing state while protecting themselves from positive feedback loops linked to the constant activation of the DNA damage response. This capacity requires changes in core cellular programs. Understanding how different cell types can undergo extensive changes in their transcriptional programs, metabolism, heterochromatin patterns, and cellular structures to induce a common cellular state is crucial to preventing cancer development/progression and to improving health during aging. In this review, we discuss how senescent cells continuously evolve after their initial proliferative arrest and highlight the unifying features that define the senescent state.

## 1. Introduction

Cellular senescence is generally an irreversible proliferative arrest in damaged normal cells that have exited the cell cycle. These cells display high metabolic activities [1], remain viable, and actively suppress apoptosis [2,3]. Senescent cells present unique morphological and molecular characteristics and functions that distinguish them from other nondividing cell populations, such as quiescent cells and terminally differentiated cells [4,5,6]. The hallmarks of cellular senescence include: prolonged cell cycle arrest, transcriptional changes, acquisition of a bioactive secretome, known as the senescence-associated secretory phenotype (SASP), macromolecular damage, and deregulated metabolism [7].

Replicative senescence was the first cellular senescence subtype to be described [8]. It is induced after serial propagation of normal human cells in culture and is caused by telomere erosion and the consequent increase in DNA lesions [9,10,11,12]. The limited lifespan of most (perhaps all) cultured primary cells is influenced by the species and tissue type from which they were derived. Senescence can also be triggered by many other intrinsic and extrinsic factors, particularly, replicative stress, oxidative damage, metabolism dysfunctions, cytokines, oncogene activation, and chemotherapy agents. All these factors can induce DNA damage and senescence in normal and cancer cells (in some contexts) [6]. Cellular senescence occurs not only *in vitro* (i.e., cell culture models), but also in various tissues *in vivo* [13,14,15,16]. 

Senescence is an important contributor to cancer and aging, two processes characterized by a time-dependent accumulation of cell damage and dysfunction. Senescence markers are detected in premalignant tumor lesions but not at later stages of tumor development [17,18,19]. The proliferative arrest imposed by cellular senescence represents an early barrier against cancer initiation by preventing the propagation of damaged DNA to the next generation of cells [18,20]. Therefore, it has been proposed that senescence escape is required for tumor progression to overt malignancy [18,21]. On the other hand, senescent fibroblasts can influence their local environment by turning into proinflammatory cells that can promote the growth of transformed or preneoplastic neighboring epithelial cells in culture and *in vivo* [22,23,24]. During aging, senescent cell accumulation in various tissues promotes chronic inflammation that accelerates age-related dysfunctions [16]. Moreover, stem cell senescence caused by telomere shortening can negatively affect tissue homeostasis and regeneration [25]. Importantly, elimination of senescent cells can promote stem cell proliferation and delay the appearance of aging features [13,14,15]. However, senescent cells are not effectively removed in aging tissues and this might reflect the age-related decline in immune functions [26]. In addition, senescent cells have been observed in many different physiological contexts and at all stages of life, with far reaching implications. Indeed, unlike chronic senescence (i.e., accumulation of deleterious senescent cells with aging), the acute induction of cellular senescence represents a transient physiological response during embryonic development [27,28] and adult tissue homeostasis (e.g., to facilitate tissue repair after liver damage, in skin fibrosis, and wound healing) [29,30,31]. 

Senescent cells are heterogeneous and perform various biological functions. Therefore, entry into senescence and the long-term cell cycle exit are regulated through different changes in gene expression, metabolism, and cell organization. Here, we describe some examples of this diversity of senescent cell types, focusing on mammalian systems, and highlight the contribution of cellular senescence to aging and cancer development. We also discuss the functions, regulation, and features of this complex and fascinating cell state. 

## 2. Senescence and the Control of Cell Cycle Arrest

Although the causes of senescence are multiple, inhibition of the cell cycle machinery is the defining characteristic and is critical for both the establishment and maintenance of all cellular senescence phenotypes. It is generally thought that the growth arrest of senescent cells occurs through a cell cycle blockade in the G1 phase to prevent DNA replication initiation in damaged cells [4,6]. Senescent cells may also stop in G2 to block mitosis in the presence of DNA damage [32] (Figure 1). As this cell cycle arrest is initiated to ensure that damaged or transformed cells do not propagate mutations, its maintenance must be tightly controlled. These steps require the action of key cell-cycle regulators that respond to extracellular and intracellular senescence-inducing signals but that display distinct, signal-dependent gene-expression patterns. 

### 2.1. Cell Cycle Exit in G1 

Cell cycle is regulated by several key factors, including cyclins and cyclin-dependent kinases (CDKs), CDK inhibitors, and the retinoblastoma tumor suppressor protein (RB). Cyclin–CDK complexes drive G1/S cell-cycle progression through RB phosphorylation that coordinates the decision between G1 growth arrest and proliferation [33,34]. Due to the critical regulatory role of these cell cycle factors, senescence signaling typically acts by decreasing cyclin–CDK activity or increasing CDK inhibitor levels. CDK regulation through the expression of CDK inhibitors, including p21 (*CDKN1A*), p15 (*CDKN2B*), and p16 (*CDKN2A*), is of special relevance in all senescence types. High p21 or p16 expression level is sufficient to induce cell cycle arrest in early-passage human fibroblasts in culture [35,36]. Rodent and human senescent cells in culture and in many tissues typically display high levels of these proteins during normal aging [37,38,39]. As primary fibroblasts divide in culture, telomere shortening activates p53 [40]. Among the p53 target genes, p21 is induced rapidly upon telomere damage. In cultured fibroblasts, p21 levels reach a peak during the final two to three passages before senescence, but it may not be essential for senescence because its expression does not persist in senescent cells. Compared with p21, p16 accumulation is relatively slow in replicatively senescent fibroblasts [38,39]. In senescent cells, p21 enforces G1 arrest by inhibitory binding to cyclin E–CDK2 and cyclin A–CDK2 complexes, thus ensuring that RB remains hypo-phosphorylated and active. Persistent activation of RB prevents G1 to S phase progression by sequestering E2F transcription factors. Ultimately, inhibition of cyclin–CDK complexes by p21 and p16 results in the persistent activation of RB, that represses cell cycle progression by sequestering E2F family members. In senescence, p16 induction could represent the major switch to RB engagement in its active form and to irreversible cell cycle arrest (discussed further below). 

**Figure 1 ijms-22-13173-f001:**
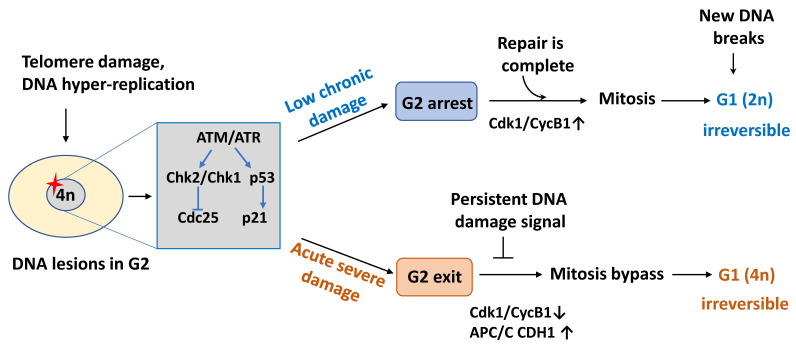
Schematic diagram of cell cycle arrest in senescent cells. In replicative senescence that results from chronic low-level macromolecular damage, cells mainly stop in the G2 phase of the cell cycle because post-replicative telomere attrition preferentially triggers the DNA damage response (DDR) at the G2/M transition. Upon damage resolution, through repair or proper post-replicative processing, checkpoint recovery allows a subset of these cells to pass through mitosis. However, at the next G1 phase, the daughter cells irreversibly exit the cell cycle in a diploid (2N DNA content) state after mitosis if new DNA breaks are generated. Conversely, a prolonged DNA damage signal emanating from severe DNA lesions leads to exit in the G2 phase, APC/C CDH1 is prematurely activated as a consequence of p21 accumulation, and these cells by-pass mitosis resulting in tetraploidy (4N DNA content) and permanent G1 arrest. CycB1, cyclin B1; APC/C, anaphase-promoting complex/cyclosome.

A decrease in the activity or levels of CDK inhibitors can lead to senescence escape, and their inactivation results in senescent cell reentry into the cell cycle. Inactivation of p21 allows human senescing fibroblasts to override senescence arrest and extends their proliferative potential, probably because in these cells CDK2 activity cannot be sufficiently reduced [41,42]. Importantly, p21 depletion ameliorates the regenerative capacity of proliferative tissues, such as the intestine and hematopoietic stem cells, in late-generation telomerase-deficient mice that experience telomere shortening and age-related degeneration [43]. Conversely, p21 inhibition in fully senescent fibroblasts leads to aberrant S phase reentry with some DNA replication, but this is not sufficient to drive these cells through a complete cell cycle [44]. Replicative senescence is bypassed upon loss of p21 [41] but not upon loss of p16 function [45,46]. This indicates that p21 has a primary role in the initial steps of the cell cycle arrest. At this stage, cells do not express senescence markers, and p21 allows them to respond to telomere damage, while maintaining the potential to recover upon telomere damage repair. This initial arrest is reversible because senescent cells that express low levels of p16 can re-enter the cell cycle upon p53 inactivation [47,48]. Over time, telomere damage signaling can promote the progression of transiently arrested cells to full senescence through p16 upregulation [48]. Nevertheless, the true contribution of p16 to the response to damaged shortened telomeres is unclear. When acute telomere damage is caused by inhibition of the telomere binding protein, TRF2, p16 protein levels increase in senescent cells [49]. In this setting, p16 responds adequately to telomere damage because p53 inactivation only partially relieves the cell cycle arrest imposed by telomere signaling. It has been proposed that active p16 signaling can compensate for p53 activity decline after full senescence establishment [49,50]. 

Telomere damage and oncogenic stress induce p16 through different mechanisms. After expression of an activated oncogene such as RASV12, p21 and p16 accumulate concomitantly and rapidly in human fibroblasts, resulting in oncogene-induced senescence (OIS) [51]. At the same time, p16 induction is robust in response to oncogenic stimuli, even in cells that undergo replicative senescence with low p16 levels [52]. In human fibroblasts that carry p16 gene inactivating mutations, senescence is bypassed [53]. Moreover, cells with low baseline p16 levels do not undergo OIS but show anchorage-independent growth (a feature of cell transformation) [54]. Benign melanocytic lesions, which are melanoma precursors in humans, express high p16 and are an *in vivo* example of OIS [19,55].

Senescence is classically viewed as a stress response but can also occur in mouse and human embryos [27,28]. The implicated molecular pathways are slightly different. In senescent embryonic cells, p21, but not p16, enforces cell cycle arrest, together with p15 that is a p16-related CDK inhibitor. Moreover, p21 is not induced by p53 or DNA damage but is regulated by the transforming growth factor (TGF)-β/SMAD and PI3K/Forkhead box O (FOXO) signaling pathways [27]. Surprisingly, developmental senescence is not required for embryogenesis because embryos lacking p21 show minimal or no developmental defects [56]. Other mechanisms, such as apoptosis, may partially compensate for the loss of developmental senescence in such embryos [27,28]. 

### 2.2. Cell Cycle Exit in G2 

Although traditionally described as an irreversible form of G1 arrest, cellular senescence can also start during the G2 phase in response to telomere damage [57], DNA damage [58], or oncogene activation [59]. Upon DNA replication completion, a large number of mitotic regulators are accumulated in healthy cells in G2 for mitotic entry and proper mitosis progression. These regulators, such as cyclin B1, are then degraded by the E3 ubiquitin ligase anaphase-promoting complex/cyclosome (APC/C) to promote mitosis exit. Their unscheduled degradation is believed to result in cell cycle exit in G2 [32]. Unlike, G1 arrest, cell cycle exit in G2 occurs rapidly, within few hours after damage [58]. Activation of p53 at G2 leads to p21 upregulation, followed by the nuclear sequestration of cyclin B1 and mitosis entry inhibition. This may appear counterintuitive, because in unperturbed cells, cyclin B1 nuclear translocation at the end of G2 triggers mitosis onset. In fact, cyclin B1–CDK1 complexes bound to p21 in the nucleus are inactive and cannot be activated upon CDK1 dephosphorylation by phosphatase CDC25C [60,61]. Consequently, these cells without cyclin B1–CDK1 activity are refractory to reactivation upon damage resolution, and they do not progress into mitosis, thus marking a point-of-no-return for the irreversible cell cycle arrest in G2 [58,62,63]. As cyclin B1–CDK1 complexes inhibit APC/C in complex with its co-activator CDH1 (APC/C CDH1) [64], the absence of mitotic CDK activity promotes the premature activation of APC/C CDH1 that targets cyclin B1 and other mitotic regulators for degradation. Moreover, p21 can indirectly activate RB family proteins that transcriptionally repress mitotic regulators, therefore stopping G2/M progression. When this mechanism fails, cells that are long-term arrested in G2 can enter a permanent G1-like state without going through mitosis (mitosis by-pass), thereby generating tetraploid cells [59] (Figure 1). Proliferation of these tetraploid cells, a potentially deleterious event, could be prevented by blocking them before the S phase, possibly through permanent inactivation of cyclin–CDK complexes and inhibition of RB phosphorylation by upregulation of p16 in G1, as discussed above. Most notably mitotic by-pass, which results from insufficient levels of mitotic proteins, is sufficient and necessary for OIS both *in vitro* and *in vivo* [59].

Therefore, G2 and G1 (for cells with 4N DNA content after mitosis by-pass) might be the phases at which senescent cells exit the cell cycle upon acute and robust DNA damage that induces a rapid cell response, typically within 24 h [58,59]. Unlike acute senescence, replicative senescence results from chronic, low-level macromolecular damage caused by telomere erosion and increased reactive oxygen species (ROS) levels (due to mitochondrial dysfunction and/or environmental insults). Upon telomere erosion, senescing human fibroblasts show a prolonged G2 arrest, indicative of some mitotic entry delay [57]. Mitotic duration increases with age, whereas mitotic transcripts decrease, leading to a higher rate of chromosome mis-segregation events during mitosis. This mitotic fidelity loss with age generates daughter cells with aneuploidy and micronuclei that ultimately lead to permanent cell cycle arrest in G1 and full senescence phenotypes [65]. This might explain the frequent occurrence of cells with two G1 nuclei, micronuclei, mitotic defects, and/or 4N DNA content in senescent cells populations *in vitro* and *in vivo* [55,59,66]. 

Altogether, different senescence stimuli engage different signaling pathways that ultimately converge to influence the levels of senescent state regulators. Therefore, senescent cells invariably display high levels of at least one of these key cell cycle inhibitors, a molecular hallmark of cellular senescence. This multiplicity of senescence markers represents different senescence stages. Some markers identify the early stage (p53 and p21), while others identify the late stage (p16 and ARF, gene products of the *CDKN2A* locus). Although frequently used, p53 and p16 are not exclusively senescence markers, but can indicate other biological functions depending on the cell context. 

## 3. Molecular Signaling and Pathways Leading to Senescence Entry

### 3.1. RB/p16 and p53 Pathways Engaged upon Senescence

Cells can become senescent in response to different damaging stimuli. Most common inducers include DNA damage (e.g., telomere shortening or dysfunction, oncogene activation, oxidative damage, radiation); metabolic stresses (e.g., mitochondrial dysfunction, redox states changes), inflammatory cytokines/chemokines and damage signals (e.g., protein misfolding, autophagy disruption). Due to the multiplicity of stress signals and effector pathways involved in its execution, cellular senescence is associated with various phenotypes the characteristics of which depend on the signals that induced senescence, the originating cell type, the time elapsed since senescence initiation, and the site where senescent cells are located. Nevertheless, some effectors are common to the majority of known senescent states. Namely, the p53/ARF and RB/p16 tumor suppressor pathways act in a cooperative and interconnected way to block the cell cycle and implement the senescence programs. The activation of only one or both pathways for the execution and maintenance of the senescence program is dictated by the type of stress signal, the tissue, and the species of origin [67], or by a compensatory response to signaling imbalances. For example, loss of RB can trigger p53 upregulation via ARF or DNA damage signaling, thereby providing a safeguard to prevent senescence escape and malignant transformation [50,68]. 

In senescent cells, p53 transcriptional activity is increased through DNA damage response (DDR)-dependent signals mainly driven by unresolved DNA damage signaling due to telomere erosion and oncogene hyperproliferation [10,11,69,70,71]. DDR can also signal to p16, although p16 may act as a secondary mechanism to mediate the senescence-related proliferative arrest following telomere dysfunction [48,49]. Cell cycle exit during OIS is strictly dependent on the activation of RB that acts downstream of p16. The p53 and RB pathways are needed for OIS initiation and also maintenance in most systems, but their specific requirement may differ in humans and mice and might be cell type-dependent. For example, mouse embryonic fibroblasts (MEFs) lacking p53 [72] but not *Ink4a (p16)* or *RB* [73,74,75] escape Ras-induced senescence. Conversely, loss of functional p16 [53] but not of p53 [51,76] enables human fibroblasts to avoid Ras-induced senescence. It appears that the signaling pathway responsible for OIS is p53-dependent in murine cells, while the RB pathway might play a predominant role in human cells. However, neither p53 nor p16 are implicated in human mammary epithelial cells [77] or melanocytes [78] undergoing Ras-induced senescence. Therefore, p53 and p16 may not be universal OIS mediators in human cells.

The regulation of the *CDKN2A* locus is complex. In proliferative cells, polycomb repressive complexes (PRC1 and PRC2) are recruited by the long noncoding RNA *ANRIL* to the *CDKN2A* locus [79]. The EZH2 histone methyltransferase (a component of the PRC2 complex) catalyzes trimethylation of lysine 27 on histone 3 (H3K27me3), a repressive mark, on chromatin to silence the *CDKN2A* locus. This epigenetic mark is recognized and enforced by PRC1 that contains B-cell-specific Moloney murine leukemia virus integration region 1 (BMI1), chromobox protein homologue 7 (CBX7), and chromobox protein homologue 8 (CBX8). Overexpression of BMI1 [80], CBX7 [81], or CBX8 [82] results in the direct repression of p16 and senescence delay in human and mouse cells. During senescence, PRC delocalization from the *CDKN2A* locus, transcriptional downregulation of EZH2 [83], and recruitment of JMJD3 [84,85], a histone lysine demethylase that catalyzes H3K27 demethylation, lead to removal of the repressive H3K27me3 mark at the *CDKN2A* locus and facilitate p16 transcription.

### 3.2. Short Telomeres and DNA Damage Signaling 

Telomere shortening is the signal for replicative senescence in human fibroblasts because telomerase reconstitutes telomere function and restores the proliferative capacity that ultimately leads to cell immortalization [8,9]. Telomeres are repetitive DNA sequences, bound by the six-subunit shelterin complex [86] that facilitates the formation of telomere loops (t-loop) to hide the telomere end [87,88], thus protecting the natural chromosome ends from signaling pathways that sense and act at DNA breaks. Shelterin allows the efficient suppression of non-homologous end joining (NHEJ) and ATM activation at chromosome ends [89]. Telomeres shorten at each cell replication because of the end-replication problem and post replicative end-processing [89,90]. This problem is solved by telomerase, the enzyme that adds back telomere repeats to compensate for this shortening. However, most somatic cells lack telomerase activity, and this progressive telomere erosion ultimately results in the loss of telomere function and triggers a permanent proliferative senescence arrest [9]. In proliferating cells, telomere shortening is not exclusively due to the end-replication problem, which would affect all telomeres equally. Indeed, random damage to telomeres through oxidative damage [91], exonucleolytic processing events, nucleases [92], secondary structures in telomeric DNA [93], and sporadic loss of large telomeric stretches [94] also can accelerate telomere shortening and dysfunction. The mean telomere length in human senescent fibroblasts ranges between 6 and 8 kbp, but senescent cells might contain a few extremely short telomeres [94] that have lost completely their protective function and expose telomere-free ends, leading to DDR activation and cell cycle arrest [10,57]. In this situation, foci of phosphorylated histone H2AX (γH2AX), which mark the presence of DNA breaks, accumulate at telomeres in human senescent fibroblasts [10,11,12]. At these foci, γH2AX rapidly concentrates and anchors DNA repair proteins, such as 53BP1, MDC1, and NBS1, in the vicinity of telomeric lesions. γH2AX focus formation correlates with the activation of the DNA damage effector kinases ATM, CHK1, and CHK2 [10,11,12]. However, as damage signals are observed also at chromosome ends that still contain significant telomeric repeats, complete disappearance of telomeric repeats is not required to signal dysfunction. A clear loss of shelterin association with telomeres has been observed in senescent fibroblasts, which is consistent with the progressive displacement of these complexes from short telomeres [57]. Similarly, acute telomere deprotection by inactivation of the shelterin protein TRF2 leads to robust telomeric ATM signaling, NHEJ-dependent telomere end-to-end fusion, and rapid cell cycle arrest [71,95]. This acute loss of telomere end protection is unlikely to mimic the events leading to replicative senescence because in senescent cells, fusions between dysfunctional sister chromatids in end-to-end joining events are more frequent than inter-chromosomal end-to-end fusions (a consequence of complete exposure of the telomeric tract to the NHEJ pathway) [57]. Therefore, too short telomeres might retain sufficient telomere length and TRF2 to suppress chromosome end-to-end fusions via the NHEJ pathway but not to efficiently protect telomeres against ATM activation. It has been proposed that an ATM-activated but NHEJ-repressed state represents an “intermediate state” of telomere end protection in cells in which TRF2 expression was partially reduced using a TRF2 mutant that cannot form t-loops [96,97,98]. However, in senescent human cells, short telomeres do not recapitulate the intermediate state phenotype predicted by this model (i.e., mild ATM activation without evidence of significant CHK2 and p53 activation) [10,11,12,57]. The activation in senescing cells of G2/M phase checkpoint markers clearly indicates that their telomeres are somehow dysfunctional, most likely after telomere losses during DNA replication [57]. How too short telomeres activate the DDR in G2 is not clear. Perhaps, severely shortened telomeres cannot fold correctly into the looped conformation that normally reforms after DNA replication and ensures that telomere ends are not mistaken for DNA breaks [99]. This suggests that t-loops cannot form below a minimum telomere length and that short telomeres may not retain sufficient length to form or stabilize t-loops. As t-loops require a G-overhang, loss of G-overhangs in senescent cells, possibly due to perturbations in end-protection, restricts t-loop formation [100]. Alternatively, in critically short telomeres, TRF2 load might become limiting to apply topological stress to the telomeric double stranded DNA. In agreement, TRF2 overexpression delays replicative senescence onset [101], indicating that TRF2 capacity to sequester the telomere terminus *via* the t-loop conformation [88] might suppress the DNA damage signal from too short telomere ends and protect them from ATM signaling. As telomeres shorten, the frequency of t-loop occurrence would decrease due to reduced occupancy by TRF2 required to form a functional t-loop structure after DNA replication. Consequently, the cell cycle stalls in G2 to provide time for the re-establishment of the folded structure after replication [57]. These cells can then progress through the cell cycle, suggesting that they reestablish a sufficient telomere protective state or that telomere fusion (although an extremely rare event) occurred between replicated sister chromatids, thus preventing further damage detection. Alternatively, cells approaching senescence may proceed to mitosis despite the presence of residual (4 to 5) γH2AX-marked unstable telomere structures [102]. Presumably, a threshold number of dysfunctional telomeres must accumulate before the signal is sufficiently high to enforce persistent DDR signaling and to force cells exit in G2 through sustained p53 activation. This threshold number is not a single DNA break because the G2/M checkpoint only responds to 10–20 DNA breaks. Therefore, cells with a low number of DNA breaks can enter mitosis [103,104]. Progression to mitosis in cells with residual damage or fused chromosomes contributes to the generation of micronuclei upon cytokinesis [105]. Inherited dysfunctional telomeres and additional DNA breaks following mitosis will cause G1 arrest in the resulting daughter cells (Figure 1).

### 3.3. Oxidative Damage and Irreparable Telomeric Lesions

Accumulation of γH2AX foci in senescent cells in culture and in cells derived from aged individuals can mark also non-repaired DNA breaks that are not necessarily the result of critically short telomeres [106]. For example, telomere dysfunction, assessed by the association of DDR proteins with telomeres, increases with age *in vivo* in baboon skin fibroblasts [107,108] and in mouse liver and gut [109]. These age-associated foci occur despite the presence of long telomeres and active telomerase in mice. In this case, telomere damage is attributed to specific features of telomeres that render them particularly susceptible to oxidation-induced damage, a type of damage that increases with aging [91,110]. Likewise, senescence of MEFs in culture is independent of telomere shortening because mouse fibroblasts have long telomeres and express the enzyme telomerase but is induced by high ambient oxygen levels (20%) that cause oxidative damage [111]. Culture of primary mouse and human fibroblasts at low physiological oxygen levels prevents senescence activation after long culturing (increasing passage number) or upon expression of activated Ras [54,111]. Conversely, telomerase expression in human fibroblasts does not protect them against senescence induced by oxidative damage [112] and OIS [113]. However, high telomerase activity prevents the activation of DNA damage signals originating from stalled replication forks inside telomeres, thus allowing OIS bypass [114]. The underlying mechanism remains to be investigated. 

The accumulation of persistent and unresolved telomeric damage might be explained by the cell inability to efficiently repair damaged telomeres [109,115]. In fact, lesions in or next to telomeres are refractory to repair as a result of TRF2 presence. TRF2 binds all along the telomeric array [116] and inhibits DSB repair completion by blocking the recruitment of ligase IV to DNA breaks that are located in or next to telomeric DNA [109,115]. Therefore, due to the accumulation of repair-resistant telomeres, the DDR is continually active, and cell cycle arrest is sustained. Conversely, DNA breaks in the rest of the genome are efficiently repaired, and DDR activation and cell cycle arrest are transient. Similarly, when cells are exposed to oxidative or DNA damaging agents, if the damage occurs in telomeric regions, TRF2 inhibits its repair and contributes to an unresolved DDR, which helps to stabilize the senescence arrest [115]. Nevertheless, it is unlikely that the inability to properly repair DNA breaks in telomeres is the only mechanism by which senescence is induced following exposure to exogenous DNA damaging agents.

### 3.4. Oncogene Activity and Replication Stress

DNA damage signaling is also involved in the initiation of stress-induced premature senescence in response to acute cellular stresses, such as oxidative damage generated by ROS [117] and oncogene activation [69,70]. Oncogenes, typified by oncogenic Ras, that deliver strong mitogenic signals engage senescence-related cell cycle arrest when expressed in normal fibroblasts in culture [51]. As the endogenous expression levels of oncogenes are unlikely to cause OIS, the stochastic accumulation of mutations might be needed for OIS induction [118]. Activated oncogenes deregulate cell cycle entry by increasing the activities of CDKs that function as positive S phase regulators. Consequently, oncogenic Ras drives the initial hyperproliferative phase that deregulates the usage of DNA replication origins, resulting in increased replication errors, DNA breaks, and DDR initiation [69,70]. Oncogenes may induce associated DNA damage by transcriptionally downregulating ribonucleotide reductase M2 (RRM2), a regulator of the nucleotide metabolic pathway that ultimately affects the metabolism and decreases dNTP levels [119]. Oxidative stress also may amplify DNA damage signaling in oncogene-overexpressing cells because high ROS levels have been detected in these cells [120]. ROS can damage DNA directly or through oxidative lesions that increase replication fork collapse by preventing its progression, thereby exacerbating the formation of DNA breaks and DDR activation [121,122]. Importantly, this arrest can be suppressed by inactivating the ATM or CHK2 kinases, supporting the causative role of DNA damage signaling in response to oncogenic activation in OIS [70]. In addition to aberrant DNA replication, unrepaired telomeres contribute to DDR signaling in OIS, without significant telomere shortening. Indeed, such telomeres are not visibly shorter than functional telomeres [114] in human melanocytic nevi with frequent *BRAF* mutations, a typical OIS feature *in vivo* [19].

A second DNA damage-independent pathway leading to p53 activation in OIS involves ARF (murine p19ARF and human p14ARF) that is encoded by a gene located in the *CDKN2A* locus and overlapping with *INK4A* [123]. ARF acts mainly by binding to MDM2, a p53-specific E3 ubiquitin ligase, to prevent its effect on p53 inactivation, thereby stabilizing and enhancing p53 activity. The separate *INK4A* and *ARF* promoters can differentially respond to input signals and can be independently silenced in tumors. In mouse models, deficiency of one or both proteins encoded by the *CDKN2A* locus results in tumor-prone animals [123]. However, the regulation of ARF expression in humans and mice differs significantly in response to oncogenic signals, possibly because of the limited homology between the mouse and human *CDKN2A* promoter sequences [124]. Notably, mouse, but not human, p19ARF is upregulated by various oncogenes (e.g., Ras and E2F), leading to p53 activation and senescence [125,126]. In humans, Ras activation appears to induce only p16 [19,127]. The two genes are largely coregulated in rodents, although some stimuli can selectively regulate p16 or ARF alone [128], whereas co-regulation of human p16 and p14ARF is uncommon [123]. For example, N-RAS- or B-RAF-activating mutations are commonly found in benign melanocytic nevi that show features of senescence, such as elevated p16 levels, but that do not express appreciable levels of ARF or p53 [19,55]. ARF might not play a role in senescence induction as a first-line defense against oncogenic events in melanocytes, whereas ARF is an important component in melanoma suppression. Indeed, its locus is more commonly deleted or silenced than mutated in melanoma, the more malignant later stage of disease [19,55]. 

As oncogenic stimuli can activate the DDR pathway and ARF, an important issue is whether these two pathways are engaged concomitantly or at different stages of cancer development. It is now acknowledged that increasing ARF levels often correlate with more advanced stages of cancer development and less frequently with pronounced DDR activation in response to persistent oncogenic activation [129]. Oncogene-induced replication stress and DNA breaks account for the early DDR activation mediated primarily by phosphorylation-dependent signaling [69]. Conversely, efficient ARF induction mainly reflects a transcription-based mechanism under continuous oncogenic activation [129]. The delayed ARF expression increase probably reflects the complex organization of the responsive elements within the ARF promoter [130,131,132]. 

## 4. Transcriptional and Post-Transcriptional Control of Senescence

Senescent cells are heterogeneous and are constantly evolving, therefore they must undergo multiple cellular and molecular changes (Figure 2). In this section, we will describe the transcriptional and post-transcriptional programs required to induce these changes.

### 4.1. The Senescence Transcriptional Program

#### 4.1.1. Senescence Core Genes 

The first and main step of the senescence transcriptional program is blocking the cell cycle. Many global gene expression analyses to identify gene expression signatures of senescence have highlighted highly variable and heterogeneous profiles in function of the cell of origin, the mode of induction, and the time after induction of the senescence program [133,134]. Similarly, single-cell transcriptomics studies revealed that senescent cell populations are composed of a mixture of cells with different mRNA expression profiles [135]. Global senescence-associated transcriptome signatures have been characterized mainly in fibroblasts. In these senescence-associated signatures, genes involved in growth factor signaling, cell cycle progression, DNA replication, and mitosis progression control are downregulated [68,136,137,138,139,140]. The downregulation of cell cycle genes is not specific to senescence because they are generally repressed when cells stop proliferating. Nevertheless, RB is uniquely required to repress the transcription of replication genes and then to stop DNA synthesis during senescence [68]. Conversely, RB family members have redundant activity in repressing these genes during quiescence [73]. Half of the genes differentially expressed by senescent fibroblasts (regardless of the senescence inducer) are not shared with quiescent cells [133]. Genes encoding factors that promote cell cycle arrest, such as CDK inhibitors, are upregulated upon senescence induction [139,140]. DNA repair and chromatin organization genes are downregulated [133,139,140]. Moreover, a group of genes encoding proteins involved in spindle assembly and chromosome segregation, such as CENP-E, are downregulated [137,140]. The downregulation of genes encoding proteins that ensure mitotic fidelity might increase aneuploidy and genomic instability. 

In addition to losing the capacity to divide, changes in gene expression alter other cellular processes and biochemical features in senescent cells [141]. For instance, in senescent cells, biosynthetic activity is decreased due to the downregulation of genes required for DNA, RNA, and protein synthesis, as well as of genes with a role in glucose and fatty acid metabolism [133]. Mitochondrial genes, particularly nuclear-encoded components of the electron transport chain [142], also are downregulated at senescence, in line with the substantial decrease in mitochondrial function and marked ROS production (discussed below). The reduction in mitochondrial activity and protein synthesis might directly lead to inhibition of cell proliferation due to a decrease in the available energy and raw materials. 

Conversely, genes involved in membrane trafficking and intercellular signaling, such as cell–cell adhesion molecules and cell–surface receptors, are upregulated in senescent cells [133,140]. These changes are consistent with the altered morphology and increased adhesion of senescent cells to the extracellular matrix (ECM), usually mediated through membrane-associated proteins. Senescence also is associated with upregulation of anti-apoptotic proteins of the BCL-2 family, particularly BCL-2, BCL-Xl, and BCL-W [14,15,143,144], downregulation of apoptotic effectors, such as caspase-3 [145], and altered p53 signaling [146] that might protect senescent cells against apoptosis (discussed below). 

#### 4.1.2. Genes Implicated in the Senescence-Associated Secretory Phenotype

In addition to the downregulation of cell cycle genes, senescent cells are programmed to secrete different chemokines, pro-inflammatory cytokines, growth factors, and matrix-remodeling enzymes that define the SASP [136,147,148]. This dynamic and complex secretory activity is one of the main feature of senescent cells. Although the SASP is a feature shared by different senescence types, its regulation is considerably heterogeneous and is influenced by the cell type and the level and exposure duration to the initial senescence inducer [149]. Time-series transcriptomic profiling studies indicate that the SASP composition changes in a temporal manner during the establishment of cellular senescence [150]. Consequently, it is regulated at multiple levels (transcription, translation, mRNA stability, and secretion) [151]. It is also clear that the SASP response relies on positive autocrine and paracrine feedback loops to provide a highly sensitive and robust mechanism of global SASP amplification [152]. 

The SASP response is dynamically regulated at the transcriptional level. Most SASP regulators converge to the CCAAT/enhancer-binding protein β (C/EBP-β) and nuclear factor kappa-B (NF-κB) transcription factors that cooperatively regulate SASP factors induction in various senescence contexts [153,154,155]. Activation of the SASP program requires a sustained DDR signal but is independent of p53, p21, and p16 (the mediators of cell cycle arrest in senescent cells) [147,148]. It has been proposed that the ATM kinase interacts with and phosphorylates the regulatory NF-κB essential modulator (NEMO) in the nucleus, contributing to NF-κB activation. These post-translational modifications lead to the nuclear export of the ATM/NEMO complex to the cytoplasm where NEMO activates the IκB kinase (IKK) α and β proteins that in turn phosphorylate the inhibitory IκB proteins. Consequently, IκB proteins are released from the complex and degraded by the proteasome. IκB degradation allows NF-κB translocation into the nucleus where it transactivates several SASP genes [156,157]. In the early stage of senescence, the production of interleukin IL-1α (IL-1α), which acts intracellularly or as a cell membrane-bound protein, initiates a feed-forward loop to strengthen C/EBPβ and NF-κB activity and amplify SASP signaling [153]. In addition, IL-1α and IL-1 receptor maintain the upregulation of IL-6 and IL-8 that in turn engage a positive feedback loop *via* amplification of C/EBPβ activation [154,158]. Considering the central role of NF-κB transcriptional activity in SASP induction, NF-κB inhibition in senescent cells selectively represses SASP genes that require this transcription factor [155].

ATM signaling does not regulate the entire production of SASP factors, although it is required for the secretion of IL-6 and IL-8, the most conserved and robustly expressed inflammatory cytokines [148]. Other major modulators that act upstream and/or synergistically with NF-κB include the stress kinase p38MAPKα and the transcription factor GATA4 [159,160]. NF-κB signaling can be activated by p38MAPKα, independent of the DDR [159]. GATA4 accumulates in senescent cells and induces the expression of IL-1α and TRAF3IP2 (an E3 ubiquitin ligase) that activate NF-κB to initiate and amplify the SASP response [160]. GATA4 induces the SASP through a distinct (p53- and p16-independent) branch of the DDR signaling pathway to facilitate senescence [160]. The SASP is regulated also by environmental factors, particularly oxygen availability, that can control the expression of SASP-related genes [149,161]. Hypoxia impairs mammalian target of rapamycin (mTOR) activity, leading to reduced IL-1α translation to support NF-κB activity and SASP induction [162,163]. In hypoxic senescent cells, mTOR activity decrease is mediated by activation of AMP-activated protein kinase (AMPK) that helps to maintain the energy balance under low-oxygen conditions [161,164]. 

The SASP can also be regulated through epigenetic mechanisms. For example, persistent DNA damage leads to proteasomal degradation of major histone H3K9 dimethyltransferases. This results in a decrease in histone H3 lysine 9 dimethylation (H3K9me2; a repressive chromatin modification) at the promoters of SASP genes that subsequently enhances IL-6 and IL-8 induction [165]. The increased expression of the H3K79 methyltransferase DOT1L during OIS promotes H3K79me2/3 occupancy at the *IL1A* locus, contributing to SASP gene expression [166]. SASP gene expression is also directly regulated by the histone variant macroH2A1 that accumulates during senescence [167]. Upregulation of SASP genes is balanced by a negative feedback loop whereby macroH2A1 activates DDR signaling that leads to macroH2A1 removal from the chromatin of SASP genes, thus reducing their expression [168]. Among the high mobility group (HMG) family members, HMGB2 is a non-histone chromatin-binding protein that remodels the chromatin architecture and binds to the loci of key SASP genes. Then, HMGB2 prevents their incorporation into transcriptionally repressive heterochromatin regions, facilitating their expression [169]. More directly, the chromatin reader BRD4 is recruited to super-enhancer elements that form adjacent to key SASP genes and is required for their expression [170].

Importantly, all these factors act in parallel and influence each other, particularly to regulate the expression of pro-inflammatory proteins within the SASP. However, the SASP composition can vary during senescence development. Particularly, NOTCH activity in OIS enables a switch from an early pro-regenerative secretome associated with TGF-β to a late secretome rich in inflammatory factors (i.e., inflammatory SASP) [150]. Mechanistically, a high level of membrane-bound NOTCH activity restrains C/EBPβ activity, leading to the negative regulation of inflammatory cytokine production. This inhibition is transient, and at later stages of senescence, NOTCH1 expression decreases, entailing the activation of C/EBPβ and the subsequent production of the inflammatory SASP [150].

In senescent cells, cytoplasmic DNA also acts as a danger signal and activates the innate immune sensing mechanisms to trigger the SASP. Extrachromosomal DNA molecules released into the cytoplasm of senescent cells recruit specifically the cytosolic DNA-sensor cyclic GMP–AMP synthase (cGAS). cGAS catalyzes the production of the second messenger cGMP that binds to stimulator of interferon genes (STING). This results in the phosphorylation of both interferon-regulatory factor 3 (IRF3) and of NF-κB transcription factors, thereby stimulating respectively the production of type I interferons and inflammatory cytokines [171,172,173,174]. Mechanistically, cytoplasmic DNA in early senescent cells can originate from micronuclei produced due to chromosome segregation errors during mitosis [175,176]. The nuclear envelope of these micronuclei is unstable and can rupture, leading to the exposure of micronuclear DNA that induces a cellular immune response *via* cGAS [171,172,173]. In non-dividing fully senescent cells, chromatin protrusions or nuclear budding from the primary nucleus might be the major route to form cytoplasmic chromatin fragments (CCFs), as a consequence of nuclear lamin B1 downregulation that causes the collapse of the nuclear envelope [105]. Similar to micronuclei, CCFs are recognized by cGAS and engage cGAS-STING signaling that is critical for the activation of NF-κB and the inflammatory SASP. Consistently, loss of cGAS and/or STING function reduces SASP factor levels in primary human cells and in mouse models of DNA damage-induced senescence [171,172,173]. Alternatively, an increase in the transcription of long-interspersed element-1 (LINE-1 or L1), the only human retro-transposable element, during senescence could partly contribute to the accumulation of DNA fragments in the cytoplasm [177]. As L1 displays high reverse transcriptase activity, its activation results in cDNA accumulation in the cytoplasm that reinforces cGAS signaling activation to produce the SASP response in senescent cells and chronic inflammation in aged mice [177,178]. Lastly, cytoplasmic mitochondrial DNA derived from dysfunctional mitochondria also can be sensed by the cGAS–STING pathway [142]. Therefore, accumulation of extranuclear DNA (CCFs, mitochondrial DNA, cDNA, and nuclear buds) activates cGAS–STING signaling to regulate SASP expression. 

### 4.2. Post-Transcriptional Regulation of Senescence

Post-transcriptional regulatory pathways also contribute to control senescence through the action of mRNA-binding proteins (RBPs) and noncoding RNAs, particularly specific microRNAs (miRNAs), the levels of which help to mediate the senescence state [179,180]. RBPs, such as human antigen R (HuR), AU-binding factor 1 (AUF1), and tristetraprolin (TTP), can directly or indirectly control the turnover and translation of mRNAs that encode senescence proteins [181,182,183]. For example, p16 mRNA stability is reduced by the RBPs hnRNP A1, hnRNP A2, and AUF1 [184,185]. Senescent cells express reduced levels of nuclear factor (NF90), an RNA-binding protein that suppresses the translation of SASP factors, such as MCP1, GROα, and IL-6 [186]. Consequently, the reduction in NF90 levels amplifies the production of several SASP factors. RBPs also are major regulators of genes involved in DDR and in the prevention of genome instability [187]; however, only few of these RBPs are functionally involved in promoting or suppressing cellular senescence [188].

Gene expression is robustly regulated at the post-transcriptional level also by miRNAs. These short noncoding RNAs (18–25 nucleotides long) repress gene expression by binding to complementary sequences on the 3′UTR of the target mRNAs and by blocking their translation and thus promoting their degradation [189,190]. A single miRNA can simultaneously regulate multiple target mRNAs, and different miRNAs might co-regulate the same mRNA, consistent with their role as regulatory molecules that fine-tune gene expression [191,192]. Importantly, miRNAs are differentially expressed during senescence and regulate key nodes of the senescence signaling pathways through direct binding to the mRNAs of p53, p16, SASP factors, and other senescence-regulatory proteins [193]. Overall, the reliance on regulatory miRNAs to regulate SASP and senescence activation allows rapid changes in mRNA stability and translation to ensure a tight control of gene expression. 

Alternative splicing also plays a role in the post-transcriptional regulation of gene expression during cell senescence: the same pre-mRNA can generate multiple transcripts, and therefore different protein isoforms. Alternative splicing greatly enhances transcriptome diversity and complexity, leading to different protein variants with possible different or modified functionality [194]. The relative abundance of splice isoforms produced from one gene tends to change in cells undergoing senescence *in vitro* and with aging [195]. Deregulated splicing with age is largely tissue- and species-specific, and many of the affected genes are implicated in mRNA regulatory processes, splicing machinery, inflammation, metabolism, and tissue regeneration [193,196]. Deregulation of the normal splicing patterns can be partially attributed to the use of alternative splice sites and to changes in exon exclusion or intron retention that can alter the protein structure, localization, regulation, and function [197]. Some of these changes, for example, in the genes encoding nuclear lamin A (*LMNA),* S-endoglin (*ENG*), p53 (*TP53*), and the EAAT2 glutamate transporter (*EAAT2*), contribute to aging-related phenotypes [198,199,200,201]. In addition to the direct changes due to alternative splicing of age-related genes, altered expression of splicing factors has also been associated with cell senescence and age-related phenotypes. For example, the level of the splicing factor SRSF3 decreases in replicatively senescent human fibroblasts, and its knockdown promotes p53-mediated senescence by directly upregulating p53β, an alternatively spliced p53 isoform [201]. The DDR might be implicated in the increased production of p53β by regulating alternative splicing and splicing factor activity [202]. Functionally, p53β is required for senescence induction, possibly through transcriptional repression [202]. Other splicing regulators such as the RNA-binding protein polypyrimidine tract binding protein 1 (PTBP1), regulate the alternative splicing of genes involved in intracellular trafficking and are required for the pro-inflammatory SASP [203]. Consequently, the imbalance in the expression of protein isoforms caused by senescence-related splicing alterations could reflect the inability of senescent cells to properly respond to cellular stress, highlighting the decline in cell adaptability and plasticity during aging. 

## 5. Changes to the Cell State during Senescence

Senescence development involves substantial changes in cell metabolism, morphology, and structures (Figure 2). Functional and molecular alterations of cell structures are associated with senescence establishment and are essential for regulating other senescence features, such as the increase in metabolic activity and protein synthesis.

### 5.1. Senescent Cell Metabolism 

Cellular metabolism changes are important for the function and fate of senescent cells [204]. Although senescent cells do not divide, they display a very active but altered metabolism, with increased glycolysis and mTOR activity [204]. The increased metabolic demands are related to their increased size, elevated production of secreted proteins (SASP), and increased oxidative stress and endoplasmic reticulum (ER) stress after cell cycle exit [205]. This results in different metabolic needs compared with proliferating cells and requires changes to support these demands. Senescent cells exhibit a shift toward elevated glycolysis with an imbalanced activity of glycolytic enzymes that results in a reduced energetic state when cell enter replicative senescence [206,207]. Increased aerobic glycolysis compensates for the reduced adenosine triphosphate (ATP) production caused by mitochondrial respiration decline during senescence [1,208,209,210]. In the early stage of senescence, mitochondria do not function properly and display impaired oxidative phosphorylation capacity and reduced inner membrane potential, resulting in ROS overproduction [120,121,208,211]. Due to their functional defects, the mass and number of mitochondria are increased in senescent fibroblasts [212]. Increased mitochondrial biogenesis is dependent on ATM-mediated activation of the Akt/mTORC1 phosphorylation cascade, leading to stimulation of the mitochondrial biogenesis regulator peroxisome proliferator-activated receptor gamma coactivator α (PGC1-α [213]. Moreover, damaged mitochondria are insensitive to mitophagy (i.e., selective autophagy of mitochondria), and consequently, mitochondrial number and size are not properly regulated in senescent cells [214]. Removal of mitochondria in senescent cells disrupts the feedforward cycle that involves ROS production and persistent DDR activation, while preserving their cell cycle arrest [215]. In these cells, SASP gene expression alteration is not caused by insufficient energy levels because ATP levels are high due to increased glycolysis. Therefore, it seems that at least in some contexts, the execution of the senescence program is compromised not by insufficient energy levels but rather by mitochondrial oxidative metabolism status. Accordingly, a metabolic shift from glycolysis towards mitochondrial oxidative respiration through activation of mitochondrial pyruvate dehydrogenase is required to establish and stabilize the OIS-associated cell growth arrest [209]. Moreover, during OIS, fatty acid metabolism is altered, glucose consumption is enhanced, and the utilization of pyruvate in the tricarboxylic acid cycle and nucleotide deficiency are increased [119,216,217,218,219]. Senescence induced by nucleotide deficiency causes aberrant DNA replication but can be overcome by ATM inactivation through restoration of glucose and glutamine consumption [220]. This supports the causative role of metabolic changes in senescence induction (Figure 3). 

Despite the increased number of mitochondria, mitochondrial dysfunction in senescent cells impairs metabolism by compromising ATP production, and also through the loss of the biosynthetic precursor pools and the inability to maintain the redox balance. This metabolic stress due to the imbalance in metabolic intermediates can be relayed through metabolic signaling that also contributes to senescence *via* multiple signaling pathways. Reduced ATP production increases the AMP-to-ATP ratio, a measure of the cell energy charge. This leads to activation of the energy sensor AMPK to coordinate activities for adapting to the metabolic stress. AMPK increases ATP levels through the activation of mitochondrial biogenesis and the stimulation of catabolic pathways, such as autophagy, induction of fatty acid oxidation, and glucose uptake [221,222]. Additionally, chronic AMPK activation promotes senescence *via* multiple mechanisms. Indeed, AMPK regulates cell cycle arrest and senescence by activating p53 that upregulates p21 transcription [223]. AMPK also prevents cytoplasmic translocation of the mRNA-stabilization factor HuR, thereby increasing p21 and p16 mRNA stability and enhancing RB activity [224]. AMPK signaling is also induced by the reduced cytosolic NAD^+^/NADH ratio as a result of mitochondrial defects [225]. In this study, mitochondrial dysfunction, induced by depletion of mitochondrial sirtuins or mitochondrial DNA or by inhibition of the electron transport chain, caused senescence without ROS hyperproduction and DDR activation. Growth arrest due to these mitochondrial perturbations was rescued by the exogenous supply of the electron acceptors pyruvate and potassium ferricyanide that artificially restore NAD^+^ levels. This identifies imbalanced NAD^+^/NADH levels as a stress signal secondary to mitochondrial dysfunction. Notably, such mitochondrial dysfunction may engage cellular senescence with a specific secretory profile that lacks the IL-1α/NF-κB-inflammatory component and that is regulated through AMPK-mediated p53 activation [225]. Therefore, mitochondrial dysfunction, initiated by pathways different from DNA damage, can determine the SASP quality because local extracellular factors, such as pyruvate, can modify the SASP in senescent cells [225].

In addition to stopping senescent cell proliferation, p53 and RB mediate senescence-related metabolic changes by balancing their effects on glycolysis: p53 restricts the glycolysis activity of senescent cells through several mechanisms [226], while RB upregulates glycolytic genes, resulting in high glycolytic activity [227]. In fully senescent cells, RB also stimulates mitochondrial oxidative phosphorylation [227]. The resulting activated metabolic flow efficiently produces metabolites and the energy molecule ATP that regulate the SASP. As the p53 response efficiency declines during aging [228], aerobic glycolysis is increased as well as the proinflammatory SASP *via* NF-κB signaling [155]. Similarly, during senescence, mTORC1 activation due to sustained DDR does not favor growth but promotes mitochondrial biogenesis, thus contributing to the ROS-dependent DDR persistence and to SASP regulation [215]. The increase in mTORC1 activity is essential for implementing the SASP by modulating the translation of IL-1α and MAP kinase-activated protein kinase 2 (MAPKAPK2) [162,163]. In turn, IL-1α activates NF-κB to trigger SASP amplification [163]. MAPKAPK2 phosphorylates the RNA-binding protein ZFP36L1, thus preventing its binding to and its ability to degrade SASP RNAs [162]. 

Although still poorly understood, chemotherapy-resistant senescent cells can engulf neighboring normal and tumor cells by phagocytosis for survival advantage [229]. Indeed, upon engulfment and processing through lysosomes of these cells, biosynthetic material and energy are released to sustain the high metabolic needs of senescent cells [229]. 

### 5.2. Cellular Structures: Membrane-Bound Organelles

Senescent cells display characteristic morphological alterations, including flattened, enlarged cell shape and an increase in focal adhesions due to CDK5-dependent activation of the cytoskeleton protein ezrin [230]. These changes reflect the increased alterations in abundance and activity of membranous organelles, particularly mitochondria, lysosomes, and ER. Although in senescent cells, the number and mass of organelles increase, these senescent organelles display functional defects and modified communication through the release of metabolites [215,231]. Consequently, to maintain homeostasis, senescent cells may produce more organelles to compensate for their declined function upon damage by mitochondrial oxidative stress. However, the newly generated organelles also may be exposed to oxidative stress and accumulate alterations, and this aggravates the senescence phenotype. The defective autophagy also can potentiate this effect through loss of its quality control capacity. 

Senescent cells are characterized by an expanded lysosomal compartment and vacuoles [231]. Lysosomes are acidic organelles that contain hydrolytic enzymes required for protein degradation in autophagy. In senescent cells, the activity of lysosomal β-galactosidase is significantly increased as a consequence of the increased lysosomal mass and becomes detectable experimentally at pH 6.0 [232]. Senescence-associated β-galactosidase (SA-β Gal) activity is one of the first and most used biomarkers of senescence, although it has limitations because it can be detected in non-senescent cells that are confluent or serum-starved [233]. Lipofuscin are lysosomal aggregates of non-degradable oxidized protein, lipid, and metal that also accumulate in senescent cells. Lipofuscin accumulation reflects reduced lysosomal activity, altered metabolism, and autophagy dysfunction. Lipofuscin aggregates can be assayed by staining with the Sudan Black B dye, and they are a sensitive biomarker of senescent cells *in vitro* and even in *in situ* samples [234,235]. 

Senescent cells also display ER expansion and biogenesis to adapt their capacity to the high amount of synthesis, maturation, and secretion of factors involved in the SASP [1,78,236,237]. ER capacity to properly synthetize proteins could be overwhelmed, thus leading to accumulation of misfolded proteins that induces a stress response in the ER. This leads to increased ROS levels that can further impair protein folding and the formation of correct disulfide bonds in many SASP proteins (this requires a controlled oxidant state and glutathione content). Oxidation or decreased expression of some chaperones and folding enzymes with age correlates with the reduction in their enzymatic activity that impairs the ER folding capacity [238]. As a consequence, progressive protein misfolding/aggregation or massive SASP protein synthesis in senescent cells cause the loss of the protein quality control homeostasis (or proteostasis) and activation of the unfolded protein response (UPR). The UPR signaling seeks to limit ER abnormal protein load and to reduce ER stress by inhibiting protein translation [239], by upregulating various ER chaperones that contribute to the correct organization of misfolded proteins [240], and by degrading misfolded proteins *via* the proteasome through an ER-associated degradation process [241]. For example, in OIS, the increased ROS production caused by ER stress leads to DNA damage that activates ATM. Active ATM triggers the removal from SASP genes of the histone variant macroH2A1, leading to inhibition of their transcription and reduction in ER stress [168]. 

### 5.3. Autophagy Regulation of Senescence

Senescence involves cellular remodeling, enlarged size, higher organelle content, and increased metabolism. These changes lead to the senescent phenotype, partly supported by mTOR activity that promotes cellular anabolism and inhibits the catabolic autophagy pathway. Macro-autophagy (hereafter autophagy) describes the bulk or selective degradation by lysosomes of damaged macromolecules (e.g., proteins) and organelles, and the recycling of degradation products to sustain cell biosynthetic and bioenergetic demands [242]. Under normal conditions, basal autophagy contributes to maintaining the metabolic homeostasis and controls the quality of cell components [243]. Stress can cause adaptive autophagy responses. Indeed, autophagy is also activated in senescing cells to limit damage by removing defective organelles that may be a source of oxidative stress and consequently DNA damage. Autophagy is also required for the execution of the senescence program because its inhibition delays the senescence-related cell cycle arrest and SASP factor accumulation, at least in OIS [244]. Functionally, autophagy is activated at a later stage than DDR signaling to sustain the bioenergetic needs and to supply metabolites for SASP factor synthesis [245]. Overall, autophagy can both promote and inhibit senescence in different contexts because it modulates several effectors with opposite functions in senescence regulation.

In senescence induced by oncogenic Ras, protein synthesis and autophagic degradation are simultaneous activated, and the same cell displays high autophagic activity and activated mTOR, the autophagy inhibitor [244]. This paradoxical outcome is possible through the spatiotemporal sub-compartmentalization of mTOR and autophagy that allows their simultaneous activation [246]. The mTOR–autophagy spatial coupling compartment (TASCC), which is close to the nucleus, is highly enriched in mTOR and keeps mTOR away from the autophagy machinery, located away from the TASCC, to avoid its inactivation. In the TASCC, mTOR is closely associated with lysosomes/autolysosomes that generate a high flux of recycled amino acids and other metabolites. These amino acids are then used by mTOR to support the increased biosynthetic demand during the acquisition of the OIS phenotype [246]. 

Senescence can be promoted by general autophagy through the TASCC. Conversely, selective autophagy, for instance of GATA4 that accumulates during senescence induction [160], can prevent senescence [160,246]. Under basal conditions, the autophagic adaptor SQSTM1/p62 interacts with GATA4 and mediates its degradation by selective autophagy. Conversely, upon irradiation or oncogene hyperactivation, activation of ATM and ATR suppresses selective autophagy by promoting GATA4 dissociation from SQSTM1/p62, resulting in GATA4 stabilization. Then, GATA4 initiates the synthesis of SASP factors *via* NF-κB activation, partly mediated by IL-1 α production [160].

OIS relies not only on the autophagic turnover of cytoplasmic material but also on the specific autophagic degradation of nuclear lamin B1 in lysosomes, where it is delivered by nucleus-to-cytoplasm transport [247]. The lipidated form of the autophagic protein LC3B, which is involved in autophagy, membrane trafficking, and substrate delivery [248], interacts with lamin B1 at the nuclear lamina. However, this interaction, does not result in lamin B1 degradation by autophagy under basal conditions. Conversely, upon oncogene activation, lamin B1 and LC3 interact with lamina-associated domains (i.e., transcriptionally inactive heterochromatin) and are extruded through nuclear blebbing into the cytoplasm, forming CCFs that are then degraded by cytoplasmic autophagy. Importantly, autophagy inhibition maintains the nuclear envelope integrity in senescent cells by suppressing lamin B1 degradation and CCFs formation [247]. Therefore, CCFs formation relies on the constitutive interaction between lamin B1 and LC3, and CCFs formation is required to reinforce OIS by initiating the SASP program [171,172]. Accordingly, CCFs clearance by autophagy represses senescence by preventing CCFs-induced cGAS/STING activation and SASP production [249]. Therefore, autophagy might be a protective pathway against micronucleus formation and CCFs accumulation. 

## 6. Ensuring the Duration of the Senescence State 

Cell viability and a generally irreversible growth arrest are two key features of cellular senescence. These features require protection against apoptosis and stability of the cell cycle exit. In most senescence models, activation of p53 and of DDR proteins initially facilitates cell cycle arrest (as previously discussed). During senescence, the DDR continues to signal through the p53 pathway [47], due to the persistent DNA damage at telomeric sites that cannot be efficiently repaired [109,115] or through the induction of positive-feedback loops that promote constant generation of short-lived repairable non-telomeric lesions [250,251]. These factors remain the main driving force in the establishment of the senescence program. Continuous DNA damage signaling is essential for cell cycle arrest maintenance because inactivation of checkpoint kinases, such as ATM [10,12], CHK2 [11], and p53 [47,48], results in escape from this arrest. The switch from transient to irreversible growth arrest involves positive feedback loops in which mitochondrial dysfunction via ROS [215,250], pro-inflammatory SASP factors [147,154,252], extensive chromatin remodeling [253], and/or nuclear lamina and chromatin degradation [105,171,247] reinforce DDR signaling (Figure 3).

### 6.1. Protection from Apoptosis during Senescence

Senescent cells do not undergo apoptosis [143] despite the presence of high and irreparable levels of DNA damage. Indeed, induction of senescence appears protective against p53-dependent apoptosis [146] and high oxidative stress [254,255]. For this, senescent cells rely on the upregulation of pro-survival pathways (e.g., BCL-2 and Ephrins) that actively inhibit apoptosis [3,14,15]. For example, inhibition of the anti-apoptotic BCL2, BCL-W, and BCL-XL proteins induces apoptosis of senescent cells and their elimination in mice [3,14]. Upregulation of the cell cycle inhibitor p21 contributes to apoptosis resistance, for example, by inhibiting p53-mediated apoptosis after DNA damage [256,257] and by preventing the cleavage of the apoptosis effector caspase-3 and the activation of the JNK pathway (both required for apoptosis) [144]. IL-6 also facilitates the senescent phenotype maintenance by inhibiting mitochondrial-mediated apoptosis and by stimulating the pro-survival activity of NF-κB in senescent cells [258]. Senescence-associated heterochromatic focus (SAHF) formation (discussed below) also promotes cell survival by protecting senescent cells against excessive DNA damage signaling during oncogenic stress [259]. In addition, increased autophagic flux promotes senescent cell survival by facilitating the degradation of damaged proteins and dysfunctional organelles [260]. Lamin B1 and a portion of chromatin-containing DNA damage markers are degraded by autophagy into the cytoplasm in senescent cells, perhaps as a way to sustain their viability [105,247]. On the other hand, autophagy inhibition leads to toxic cell waste accumulation and elevated ROS levels that can induce senescence [261]. Likewise, the reduction in stem cell pools in aged muscles is linked to impaired autophagy that promotes senescence and proteostasis loss, increased dysfunctional mitochondria, and elevated ROS levels in these cells. Conversely, restoring autophagy activity in old satellite cells clears protein aggregates and reverses senescence [262]. Nevertheless, a balanced autophagy level in senescent cells is required to prevent autophagy-dependent cell death. 

Senescent cells can be cleared by the immune system [27,263,264] rather than directly through apoptosis. Indeed, senescent cells formed following tissue damage can promote their own clearance by secreting chemo-attractants that recruit and activate immune cells [265]. For example, upregulation of secretory factors in senescent hepatic stellate cells during liver damage facilitated their elimination by macrophages. Conversely, cells that could not become senescent due to p53 deletion were not targeted by macrophages [266]. Therefore, if senescent cells are not efficiently eliminated by the immune system, their pro-survival phenotype promotes their persistence within tissues. 

### 6.2. Positive Feedback Loops 

#### 6.2.1. Mitochondrial Dysfunction and ROS Production

The escalating ROS production by dysfunctional mitochondria aggravates nuclear DNA damage and stabilizes the chronic DDR activation, contributing to the maintenance of the pro-inflammatory and pro-oxidant senescent phenotype upon DNA damage [215,250]. Conversely, culturing cells in low oxygen or in the presence of antioxidants delays senescence onset [111,212,267,268]. Mitochondrial dysfunction is the main cause of elevated ROS production in senescence. Indeed, the selective and specific removal of mitochondria in senescent cells normalizes ROS levels [215]. Mitochondrial DNA is highly prone to oxidative damage because it is located close to the ROS generation site, and it is devoid of histones. In turn, damaged mitochondrial DNA alters oxidative phosphorylation reactions and can further enhance ROS production [269]. Downstream of DDR effectors, p16-Protein Kinase C delta (PKC delta) and p21-p38MAPK-TGF-β signaling also contribute to ROS production [250,268]. p16 cooperates with mitogenic signals to induce ROS that directly activate PKC delta to further sustain ROS production [268]. DDR signaling *via* p21 promotes p38MAPK activation and TGF-β signaling, leading to enhanced ROS generation that causes more DNA damage [250]. Moreover, ROS can be directly transferred between cells through gap junctions or released through hydrogen peroxide into the environment, where they can induce senescence in surrounding cells [270]. 

#### 6.2.2. SASP Factors 

SASP factor production requires continuous DNA damage signaling [148]. In turn, some SASP factors that strongly accumulate in the secreting senescent cells reinforce the senescence signaling in an autocrine fashion [154,252]. Specifically, the chemokine IL-8 and other chemokine receptor 2 (CXCR2) ligands strengthen growth arrest in senescent cells by activating a self-amplifying secretory loop through their ability to boost DDR signaling. Senescence induction following CXCR2 engagement is mediated by p53 and RB activation because CXCR2 depletion is sufficient for bypassing senescence [153]. Moreover, cell surface-bound IL-1 α and its receptors can sustain senescence by maintaining the high IL-6 and IL-8 secretion [158]. Other secreted proteins, such as IGFBP7 that influences MAPK signaling [271] and PAI-1 [272] that regulates the phosphoinositide 3-kinase (PI3K) pathway, also contribute to senescence stabilization by inhibiting proliferative and mitogenic pathways. In response to senesce-inducing stimuli, senescent cells produce, in a cGAS-dependent manner, type I interferons that have pro-senescence and anti-proliferative activities. Indeed, β-interferons increase ROS production, leading to DNA damage and p53 activation [273,274]. Overall, these autocrine loops lock the cell into the senescence phenotype. 

### 6.3. Epigenetic Profile and Chromatin Remodeling 

Senescent cells undergo extensive epigenome and chromatin organization changes that contribute to the persistent proliferative arrest and progression to full senescence (Figure 2). Chromatin remodeling occurs at various levels and might increase the access to DNA that is normally tightly packed, while restricting the access to other chromatin regions that are normally open. These chromatin rearrangements affect the transcriptional program of senescent cells and contribute to their transcriptome diversity [133,136,275]. 

Senescence is accompanied by global changes in the chromatin structure with the formation of compact heterochromatic regions, known as SAHF. These foci contain the condensed chromatin of one chromosome and can be observed under a microscope after DAPI staining. They are enriched in repressive chromatin marks, such as trimethylation of lysine 9 on histone H3 (H3K9me3) and heterochromatin protein 1 (HP1) [253,276]. Other proteins implicated in SAHF formation include components of the p16/RB pathway [253], high-mobility group A (HMGA) chromatin architectural proteins [277], and the histone co-chaperones HIRA and ASF1A [167,276]. Heterochromatin incorporated into SAHF results from the redistribution of pre-existing repressive heterochromatin marks rather than newly formed heterochromatin on genomic regions [167,253,278,279]. SAHF formation is strongly correlated with the loss of lamin B1 [279,280,281] that allows the redistribution and relocation of repressive histone marks from the nuclear periphery into SAHF. However, it is not sufficient for SAHF formation [279]. Reduced lamin B1 levels are observed in many senescence models, and its depletion causes senescence [279,280,281]. However, SAHF formation is not a universal feature of senescence. SAHFs are formed most commonly in OIS. Conversely, they are relatively infrequent during replicative senescence and are not observed in human tissues displaying features of senescence [52,259]. Importantly, SAHF precise function in senescence is still uncertain. It was originally proposed that heterochromatin repackaging into SAHF transcriptionally sequesters proliferation-driving genes, such as RB/E2F target genes, and reinforces the growth arrest irreversibility [167,253,276]. However, SAHF function is not limited to gene silencing, and SAHF structures may preserve the proliferative capacity of oncogene-induced senescent cells by restraining DDR signaling [259].

Another chromatin feature of senescent cells is the distention of peri/centromeric α-satellite and satellite II sequences (heterochromatic regions that are normally constitutively repressed) [282]. The distension of centromeres is termed senescence-associated distention of satellites (SADS). Unlike SAHF formation, SADS formation is conserved in different cell and senescence types [282]. SADS formation is an early event during senescence and precedes other nuclear changes, including nuclear enlargement and SAHF formation [282]. SADS formation may be linked to the decreased DNA methylation of heterochromatic regions with repetitive sequences and the increased expression of pericentric satellite DNA observed during senescence [283,284]. The early expression of pericentric satellite transcripts might contribute to strengthen the senescence-related cell cycle arrest through induction of genomic instability and DNA damage [285]. 

Epigenetic alterations, such as DNA methylation and chromatin accessibility changes, also occur during senescence. In replicative senescence, DNA in gene-poor, late-replicating genome regions and in heterochromatic regions becomes globally hypomethylated, whereas focal hypermethylation is increased at CpG islands [283]. Moreover, DNA accessibility, a marker of open chromatin, globally increases with focal declines during replicative senescence [286]. Specifically, chromatin accessibility in euchromatic regions decreases. Conversely, the chromatin of repetitive sequences (e.g., retrotransposons and pericentromeric satellite sequences) that is highly condensed in normal cells becomes more accessible in senescent cells. The opening of these heterochromatic regions in late senescent cells is associated with transcription of transposable elements (such as LINE1 elements) that can engage active transposition, potentially causing genomic instability. Therefore, some of these chromatin changes might contribute to the gene expression profile characteristic of senescent cells and further stabilize the proliferative arrest [177,287].

## 7. Relevance of Senescence in Cancer Development and Aging

Cellular senescence contributes to tissue homeostasis in many different biological processes. Senescence can be beneficial or detrimental for the organism, in function of the physiological context (Figure 4). Senescent cells play beneficial roles during embryo development, tissue repair/regeneration, and in the protection against cancer [288]. Developmental senescence is induced by developmental cues to regulate cell proliferation in embryonic structures and to induce tissue remodeling signals for proper embryo formation [27,28]. Cellular senescence optimizes tissue remodeling by promoting ECM deposition [30,31] and by inducing the plasticity of neighboring cell populations [289] that are important for mediating the balance between healing and fibrosis in wound closure. In addition, the permanent exit from the cell cycle of senescent cells prevents the propagation of premalignant cells in the context of tumorigenesis. These biological functions of senescent cells largely rely on their ability to communicate with the environment through various intercellular communication processes, including but not limited to the SASP factors, and to stimulate immune surveillance [149,290]. Timely clearance of senescent cells is essential to eliminate SASP factors and for the successful restoration of tissue function [289]. The full benefit of senescence is obtained when the presence of senescent cells is limited in time to avoid a negative outcome. Indeed, when senescent cells are not efficiently cleared and accumulate in tissues (for instance in the presence of continuous damage), the continued production of SASP factors can contribute to local inflammation and to the chronic inflammatory milieu, *via* paracrine and systemic SASP that aggravate tissue dysfunction [7,149] (Figure 4). 

### 7.1. Senescent Cells in Aging

The aberrant accumulation of senescent cells, possibly due to decreased clearance and/or chronic induction, can lead to pathology, as exemplified in aging. Aging is a progressive degenerative state accompanied by loss of tissue homeostasis, decreased regenerative capacity, deterioration of the overall organs function, and increased risk of developing age-associated diseases, such as Alzheimer’s disease, cardiovascular diseases, and cancer [291]. The connection between cellular senescence and aging is supported by the observation that senescent cells accumulate in various tissues during aging, particularly in association with age-related dysfunctions [141,292]. For example, senescent cell burden is higher in adipose tissue of elderly women with frailty and physical dysfunction than in healthier elderly women [293]. Senescent cells also accumulate at sites of age-related chronic diseases, even in younger individuals, and transplantation of a relatively small number of senescent cells accelerates aging in healthy younger mice [294,295]. Importantly, reducing senescent cell burden in mice alleviates features of aging, reduces frailty, brings health benefits, and increases the lifespan of old animals [13,16,295,296,297]. However, some senescent cells have important structural and functional roles, and the removal of non-replaceable senescent cells in the liver actually shortens the lifespan of mice [298].

Cellular senescence drives tissue aging and associated disorders by limiting the regenerative potential of stem cell pools and undifferentiated progenitor cells and by increasing chronic inflammation, ECM degradation, and metabolic dysfunction [26,288,299]. As previously discussed, these cell and tissue changes reflect the decline in mitochondrial function, loss of proteostasis, altered intercellular communication, deregulated nutrient sensing, epigenetic profile changes, and defects in DNA repair that lead to genomic instability and damage, including telomere dysfunction [299]. 

Historically, the discovery that telomere erosion acts as a “mitotic clock” at the cellular level [9] led to the hypothesis that telomere erosion and replicative senescence could concomitantly contribute to organismal aging. Transgenic mouse models without or with inducible telomerase [300,301] allowed establishing a link between short telomeres and the onset of aging phenotypes [300,302,303], progeroid syndromes [304], and chronic inflammatory and degenerative conditions [305,306]. Reactivation of endogenous telomerase reverses tissue degeneration in mice with telomere dysfunction [301]. To unveil the link between progressive telomere shortening and the onset of the human aging phenotype, telomere length in function of age has been extensively studied in blood cells (due to their easy accessibility) and in various tissues. A consistent mean telomere length shortening and changes in the abundance of short telomeres over time correlate with aging in humans [307,308]. Most tissues, both highly and slowly proliferative, show a decrease in telomere length associated with age [309,310]. However, it is not clear whether this erosion leads to a telomere length short enough to trigger senescence. In addition to telomere erosion, increased oxidative damage, DNA damaging agents, and metabolic changes also may induce cellular senescence in proliferative tissues during normal aging. This DNA damage that occurs independently of telomere shortening may be localized at telomeres [109,115], and it is based on some form of sporadic damage that might contribute to the overall tissue dysfunction and aging process [13,296]. Indeed, the frequency of these stress-induced senescent cells increases with age [108], and the rate of senescent cell accumulation in some tissues quantitatively predicts the lifespan in mouse strains [251]. Biomarkers of senescent cells have been identified in different organisms and in a wide range of tissues *in vivo**,* from tissues with a high proliferative index to non-replicating tissues. For example, senescence markers have been detected in stem cells and somatic cells of aged mice [311] as well as in skin of old baboons and humans [107,232]. Non-dividing (post-mitotic) cell types, such as neurons and cardiomyocytes, also can display senescence features. An age-dependent accumulation of neurons that expresses various senescence markers (i.e., high ROS production and oxidative damage, increased IL-6 production, chromatin reorganization, and SA-β-Gal activity) occurs in normally aging mice [312]. The role of neurons in the post-mitotic senescence state is not clear. An increase in senescent postmitotic neurons in elderly humans may contribute to the pathogenesis of neurodegeneration, cognitive decline, and dementia or Alzheimer’s disease [313]. Like in other tissues, senescence markers are enriched also in post-mitotic cardiomyocytes in aging mice [314] and could be associated with diminished cardiac function. Although replicative-associated senescence is very rare in neurons and cardiomyocytes, post-mitotic cells might become senescent possibly due to random DNA damage and persistent DNA damage signaling [312] because these cells are terminally differentiated and have already exited the cell cycle. Despite the established importance of senescence in the aging phenotypes, the identification and evaluation of senescent cells *in vivo* are restricted by the lack of reliable markers. None of the current markers used to identity senescent cells in culture are specific to cellular senescence [205]. Therefore, the role of senescent cells in aging, *in vivo*, can be studied only using a combination of several markers [315].

#### 7.1.1. Causal Role of Senescent Cells in Aging and Age-related Diseases *In Vivo*

Direct evidences that senescent cells contribute to age-related tissue dysfunction started to emerge from a transgenic mouse model known as *INK-ATTAC* in which p16-expressing senescent cells can be specifically eliminated by apoptosis [296]. When this transgenic model is bred in a progeroid mouse genetic background, clearance of p16-positive senescent cells decreases the age-associated dysfunctions and attenuates progression of aged-related disorders [296]. In addition, elimination of senescent cells from naturally aged *INK-ATTAC* mice extends their healthy lifespan and delays age-associated deterioration of several organs [13]. The physiological accumulation of senescent cells at sites of age-associated pathologies could contribute to the pathogenesis of these diseases. Other studies in mice have demonstrated the contribution of senescent cells to the pathogenesis of age-associated diseases, such as atherosclerosis, cardiovascular diseases, frailty, and osteoarthritis [3,14,15,295,316,317]. For example, in advanced atherosclerosis, plaques contain cells harboring senescence markers, and their clearance leads to a reduction in plaque number and size [316]. The elimination of naturally occurring p16-expressing cells in the joints of naturally aged *INK-ATTAC* transgenic mice decreases age-associated cartilage degradation [317]. Clearance of senescent cells with senolytic drugs (i.e., pharmacological agents that selectively kill senescent cells) improves cardiovascular function in old mice and extends the healthy lifespan of *Ercc1*^−/Δ^ mice that display progeroid features [3]. Similarly, depletion of senescent cells in normally aged mice after treatment with a senolytic drug rejuvenates hematopoietic stem cells and muscle stem cells [14]. Clearance of senescent cells, using senolytic drugs or in *INK-ATTAC* transgenic mice, reduces physical dysfunction and extends the lifespan of old mice [295]. Such studies confirm that senescent cells are a typical feature and a contributor to aging and age-related disorders, but the underlying mechanisms are incompletely understood. These effects might be mediated through reduction in the regenerative capacity of progenitor and stem cells or through the SASP detrimental effects on the microenvironment, as summarized below.

#### 7.1.2. Mechanisms by Which Senescent Cells Drive the Pathogenesis of Age-Associated Diseases

Permanent cell cycle arrest of senescent stem cells and progenitor cells leads to their exhaustion, and this can directly impair tissue maintenance, function, and regeneration. For example, age-related increased p16 expression in hematopoietic stem cells, central nervous system, and pancreatic islets is associated with a decrease in self-renewal capacity that is partly improved by p16 inhibition [292,318,319]. Regeneration of skeletal muscle relies on stem cells that remain quiescent until needed for tissue repair. In old mice, these cells switch to a senescence state caused by p16 expression and lose their self-renewal capacity. Inhibition of p16 is enough to restore muscle stem cells self-renewal capacity and muscle regeneration in old mice [320]. The SASP also could compromise cell renewal and organ regeneration/function [321,322]. For example, chronic exposure to IL-1 α, a key mediator of SASP activation, reduces hemopoietic stem cell renewal [322]. Similarly, secretion of activin A by senescent fat progenitors inhibits adipogenesis [321]. In humans, senescent fibroblasts accumulate in the lungs of patients with idiopathic pulmonary fibrosis, a progressive fatal lung disease whose incidence and severity increase with advanced age [323]. A pilot clinical study reported that senolytic treatments have therapeutic effects in patients with idiopathic pulmonary fibrosis [324]. *In vitro*, the secretome of senescent lung fibroblasts induces fibrogenesis in a paracrine manner in healthy fibroblasts. In a mouse model of lung fibrosis, depletion of senescent cells improves lung function, even if fibrosis is still visible, indicating that the SASP may contribute to fibrosis [325].

Moreover, a systemic chronic low-level inflammation, termed “inflammaging”, may underlie aging and most age-related pathologies [326]. Particularly, inflammatory cytokines, such as IL-1 α, IL-6, and TNF-α are SASP constituents associated with inflammaging and aging [327,328]. Mice models were the SASP is specifically targeted and modulated are ideal for understanding the SASP causal impact on aging, aged-related functional decline, and chronic diseases. For example, accumulation of senescent cells in *Nfkb1^-/-^* mice (a mouse model of chronic low-level inflammation) is associated with increased inflammation, decreased tissue regeneration, and accelerated aging. In these mice, anti-inflammatory treatments rescue the tissue regenerative capacity [251]. Moreover, conditioned medium from senescent adipocytes induces inflammation in healthy adipose tissue. In old mice, SASP inhibition reduces adipose tissue and systemic inflammation and enhances physical function [329]. Downregulation of the interferon-related response in aged mice reduces inflammaging in various tissues and improves aging-related phenotypes (e.g., kidney glomerulosclerosis and skeletal muscle atrophy) [177]. Similarly, inhibition of the pro-inflammatory secretome of senescent cells with a JAK inhibitor results in higher bone mass and strength in old mice [294]. Altogether, these studies highlight the SASP contribution to inflammaging and aged-associated pathologies, at least in mice. A major question is how senescent cells can have such a detrimental impact, although they represent a minor cell fraction in tissues. In addition to their contribution to inflammaging, senescent cells can spread senescence to neighboring cells (paracrine activity) *via* their SASP factors *in vitro* [252,270] and potentially *in vivo* [330]. For example, human senescent cells transplanted in muscle or skin of mice spread senescence to neighboring cells [330]. Similarly, transplantation of senescent cells in young mice induced senescence in neighboring cells and caused physical dysfunction [295]. Senescent cells would achieve this effect through locally secreted SASP factors [252]. Secreted IL-1 and TGF-β induce paracrine senescence through a mechanism that generates ROS and DDR signaling in adjacent cells [252,331]. The underlying mechanism of paracrine signaling remains to be investigated.

An excessive accumulation of senescent cells seems to increase the risk of more severe outcomes or complications in patients with coronavirus disease 2019 (COVID-19), an acute respiratory disease caused by severe acute respiratory syndrome coronavirus 2 (SARS-CoV-2). Indeed, COVID-19-related mortality is higher in older people, particularly those with chronic diseases [332]. Disease severity correlates with an increase in senescence markers in airway epithelial cells and higher levels of SASP factors in serum [333,334,335]. The biological links between SARS-CoV-2 infection and aging are not known yet, but the interactions between senescent cells, their SASP, and immune cells might be implicated in COVID-19 morbidity and mortality [290]. SARS-CoV-2 infection of lung and nasal epithelial cells *in vitro* induces cellular senescence with a strong pro-inflammatory phenotype [333,334]. In older patients, SARS-CoV-2 infection can substantially amplify the SASP of existing senescent cells that exacerbates systemic inflammation and drives senescence paracrine effects. Increased secretion of SASP factors can impair the immune system response. Moreover, the inefficient clearance of infected senescent cells could contribute to the formation of a hyper-inflammatory environment and to multi-organ damage [336]. Reducing senescent cell burden in aged mice decreases mortality after infection by a related mouse β-coronavirus [337], suggesting that cellular senescence is an important molecular mechanism of severe COVID-19. The prolonged survival of senescent cells in patients who recovered from a SARS-CoV-2 infection may contribute to increasing the risk for long-term COVID-19 symptoms in older patients [338]. 

#### 7.1.3. Inefficient Immune Clearance of Senescent Cells

It is not clear why senescent cells accumulate in many tissues during aging. As part of the wound healing or tissue repair response after an initial acute insult, senescent cells can attract and activate, through their secretome, immune cells that contribute to their subsequent removal [290]. Senescent cells can also become immunogenic by expressing surface markers that allow their specific recognition and elimination by immune cells [339]. The increased number of senescent cells with age might coincide with the decreased capacity of the immune system to recognize and/or clear them. Alternatively, immune clearance could become overwhelmed, allowing the accumulation of senescent cells. Moreover, the associated sustained inflammation could suppress immune cell function in the long term [290]. Indeed, in a context of impaired immune cytotoxicity, senescent cells accumulate with age in mice, and this is accompanied by increased inflammation and accelerated aging [340]. However, senescent cells can persist for years also in young mice [341,342], in young women who underwent chemotherapy for breast cancer [343], and in children who develop senescent cells in benign melanocytic nevi [19]. In these cases, the persistence of senescent cells might be mediated through the attenuation of the SASP proinflammatory properties because the SASP nature is influenced by the cell type of origin and senescence stimulus [133], or through changes in the SASP composition over time [150]. Alternatively, senescent cells can bypass recognition and evade the immune clearance through upregulation of human leukocyte antigen (HLA)-E ligand that can inhibit CD8+ T cell and natural killer cell function [344]. Therefore, understanding how senescent cells escape immune clearance could help to develop therapeutic interventions that eliminate senescent cells in order to alleviate age-related diseases. 

### 7.2. Senescent Cells in Cancer

Cellular senescence plays important but contrasting roles in different steps of tumorigenesis, such as tumor initiation, establishment, and escape. Under some conditions, it represents a potent tumor-suppressive barrier by blocking the proliferation of damaged cells. In other settings, senescent cells may facilitate cancer progression.

#### 7.2.1. Cellular Senescence as a Barrier to Tumorigenesis

The senescence program can be activated in normal, pre-neoplastic, and malignant cells in response to a wide variety of stimuli. Critically short, uncapped telomeres [9], oncogene stresses [51], and mitochondrial dysfunction [225] result in proliferative stress and senescence induction, thereby halting the proliferation of cells harboring mutations or genomic instability. The proliferative arrest of preneoplastic cells represents a protective barrier because cancer cells must replicate to produce a macroscopic tumor. Unlimited proliferation is the main means of acquiring oncogenic mutations and fixing the successive genomic alterations that drive clonal expansion and cancer progression [345]. Critically shortened telomeres are frequently observed in early neoplastic lesions [86,346,347], and too short telomeres can induce senescence. Loss of p53 and RB bypassed senescence in cultured human fibroblasts, and allowed entry into crisis, which is considered as a second barrier to cancer formation [348]. This extended proliferative period exacerbates telomere shortening and leads to chromosomal fusions and anaphase bridges. These fused chromosomes can initiate bridge–breakage–fusion cycles, causing more chromosome rearrangements and genomic instability that promote mitotic crisis [349,350]. This mitotic crisis in culture leads to the death of most cells but also produces the very rare immortal cell that has acquired telomerase activity and extensive gene copy number alterations [351,352]. Almost all cancer cells show defects in senescence-controlling signaling pathways downstream of telomere erosion (i.e., the p53 and RB pathways) that allow them to proliferate to the point of telomere crisis. Short telomeres and a sharp increase in genome instability are observed in the early stages of breast cancer before telomerase activation and genome stabilization [353]. Consistent with an initial period of telomere function loss during tumor development, studies on telomere dynamics and karyotype analysis in many human tumor types revealed telomeric fusions, indicative of telomere crisis [347,352,354,355]. These observations indicate that malignant progression requires a period similar to the telomere crisis that can result in large-scale genome rearrangements of the kind commonly seen in tumors [349,351]. Thus, telomere shortening may restrict and also promote tumor initiation. 

Replication stress induced by some oncogenes is a second stringent tumor suppressive mechanism in which the persistent DDR signal could cause senescence. DDR activation and senescence, presumably owing to unscheduled DNA replication, are frequently observed in the early stages of cancer lesions, [69,70,356,357]. For example, senescence occurs during the formation of benign cutaneous melanocytic nevi that express an oncogenic form of BRAF [19]. Mechanisms to evade senescence are required for malignant progression of melanocytes and involve additional mutations to prevent or bypass OIS, further underscoring the importance of senescence in counteracting tumor development. 

In mouse models, inactivating the senescence machinery results in the acceleration of tumor development, while restoration of senescence in growing tumors causes their regression consistent with the role of senescence in suppressing cancer [266,358].

#### 7.2.2. Dual Role of SASP in Cancer Pathogenesis 

Senescent cells communicate with and influence the behavior of neighboring cells, partly through SASP factors. This paracrine signaling is central to the tumor-suppressive and tumor-promoting consequences of the senescence response. The SASP composition and strength are context-dependent and highly heterogenous. SASP effects are beneficial or detrimental depending on the cell type, senescence stimulus, and the neighboring environment [24]. Many SASP factors exert tumor-suppressive activities through autocrine and paracrine signaling cascades that enforce senescence cell cycle exit and transmit senescence to different neighboring cell types [153,154,263,271,272,358]. The SASP can also exert non-cell autonomous tumor suppression by attracting and activating immune cells to actively promote the innate and adaptive anti-tumor immune response. Recruitment and activation of T cells and natural killer cells at the location of senescent cells in the tumor microenvironment or altered polarization of macrophages promote the removal of senescent or damaged cells [29,358,359,360]. In a mouse model of hepatocellular carcinoma, cancer development is kept under control by the immune system capacity to specifically identify and eliminate senescent pre-malignant cells. Specifically, secretion of cytokines by pre-malignant hepatocytes undergoing senescence upon oncogenic Ras expression resulted in their clearance mediated by CD4^+^T cells [263]. Conversely, impaired immune clearance of pre-malignant senescent hepatocytes in severe combined immune-compromised mice promoted liver cancer development [263]. 

Paradoxically, SASP factor expression by senescent cells can also facilitate cancer progression by promoting growth of pre-neoplastic cells and by modifying the tumor microenvironment. Conditioned medium from senescent fibroblasts promotes growth of pre-malignant and malignant breast epithelial cells, prostate epithelial cells, keratinocytes, and melanocytes but not normal cells [22,147]. ECM remodeling through the activity of ECM-degrading proteases can relax the structure of the tumor microenvironment, potentially promoting tumor cell motility and invasion, and consequently metastasis formation. In addition to facilitating the invasiveness of epithelial cell types through the secretion of chemokines and matrix-degrading proteases, senescent stromal cells can induce an epithelial-to-mesenchymal transition of neighboring cells, a major mechanism of tumor progression [23]. In cell xenografts experiments, matrix metalloproteinases secreted by senescent human fibroblasts increase tumor growth [361], probably by disrupting the immunosurveillance by paracrine action in the microenvironment. *In vivo*, the SASP of senescent hepatocytes accelerates tumor growth in mice in the later stages of liver disease. Moreover, peritumoral tissue senescence has been associated with poor survival in patients with hepatocellular carcinoma [362]. Conversely, SASP inhibition by the mTOR inhibitor rapamycin can suppress the capacity of senescent fibroblasts to stimulate prostate tumor growth in mice [163].

#### 7.2.3. Therapy-Induced Senescence of Cancer Cells 

Senescent cells can also arise within a tumor following anticancer treatments. Most chemotherapeutic drugs are designed to eliminate tumor cells by inducing apoptosis, but some tumor cells can enter senescence in response to DNA damaging therapies [363,364]. For example, senescence markers have been detected in human breast cancer samples from patients treated with chemotherapy [343]. Following drug-induced DNA damage, tumor cells are forced into senescence, they stop dividing, and cancer growth might be blocked. Additionally, the secretome of directly affected tumor cells or adjacent senescent stromal fibroblasts can induce paracrine senescence of nearby tumor cells [270] and can recruit immune cells to contribute to cancer cell removal, reinforcing the tumor suppressive role of senescent tumor cells [263,266]. However, the secretome of senescent tumor cells can also drive tumor progression by promoting inflammation and potentially stimulating the growth and invasiveness of nearby tumor cells [148,154,163]. It also impairs the elimination of senescent tumor cells from the immune system. Consequently, persistence of senescent tumor cells can be a source of chronic inflammation and of resistance to chemotherapy [363]. Another major issue of therapy induced-senescence is that unlike normal cells, senescence of tumor cells can be incomplete because cell cycle exit might not be irreversible in these cells [365]. The development of a tumor cell population that escapes from senescence and restarts proliferating may contribute to cancer relapse [342]. In addition, tumor cells that escaped senescence after chemotherapy display stem cell characteristics and acquire a higher tumor initiation potential *in vivo* [366]. 

## 8. Concluding Remarks

Acute senescence induction contributes to adequate tissue patterning during embryogenesis, limits tissue damage, and facilitates tissue repair to ensure organismal fitness. Conversely, accumulation of senescent cells during aging creates a pro-inflammatory microenvironment that promotes the progressive decline in tissue function. From a cancer biology perspective, the intrinsic tumor suppressive properties of senescent cells are clearly advantageous. Moreover, in senescent cells, continuous DDR signaling stimulates an inflammatory secretome that can act extrinsically to drive neoplastic progression of premalignant cells. However, SASP factors also exert a tumor suppression activity by promoting the recruitment and instruction of immune cells that contribute to the elimination of damaged cells [290]. Therefore, many questions remain on the significance (and involvement) of cellular senescence in tumor initiation, establishment, and progression; its contribution to tumor heterogeneity and resistance to therapy; and the extent to which senescent cells are necessary for normal organ function/homeostasis and whether they remain irreversibly cell cycle arrested in aging tissues. 

An important limitation in studying senescent cells is the lack of specific biomarkers to properly identify and follow them. Reliable markers to accurately distinguish senescent cells from other nondividing cell types would allow determining whether similar stages as those described *in vitro* (early to late stage of senescence) truly drive tissue dysfunction or cancer progression *in vivo*. It would also help to find ways of promoting the benefits of senescence, while avoiding the damaging consequences. It also might open new clinical opportunities by allowing the fine tuning of senescence dynamic features to improve tumor ablation, slow the loss of irreplaceable cells, or optimize the metabolism of senescent cells to regulate their function and survival. 

Much of the current knowledge on the senescent cell state comes from work performed in cultured cells. Although tissue culture systems have been crucial for understanding cellular senescence, they do not recapitulate the complexity found in an organism, for instance the multitude of signals from the microenvironment. Thus, some of the findings highlighted in this review still need to be confirmed in *in vivo* systems, such as whole animals. For example, the findings on the heterogeneity of gene expression in cultured senescent fibroblasts should be reevaluated in an organism. Senescent cell populations are generally highly heterogeneous in function of the cell type, the senescence-initiating insult, and the interactions within the microenvironment [133,134,135]. Therefore, it is reasonable to question their exact role in different tissues and under different conditions. However, it is still challenging to study how senescent cells are induced *in vivo*, how the SASP changes over time, whether/when the SASP acquires damaging traits, and how senescent cell clearance is regulated in physiological contexts. Clearly, the development of methods to isolate and characterize senescent cells from aged and diseased tissues will greatly enhance our understanding of how these cells promote tissue deterioration. 

A better understanding of the molecular pathways underlying the heterogeneity and dynamic nature of senescent cells holds promise for therapeutic applications. Cancer cells in which the molecular pathways leading to senescence are not affected respond to chemotherapy by entering senescence, which leads to tumor regression through immune clearance of senescent cells. Therapy-induced senescence of tumor cells has been proposed as an alternative and effective way of improving cancer treatment [365]. The main advantage of the senescence response caused by traditional chemotherapeutic agents is that cancer cell senescence can be induced with lower doses than those used to cause cancer cell death, thus possibly reducing the adverse side effects [363,367]. Senescence-inducing therapies rely on limiting cancer cell proliferation and also fibrosis formation during carcinogenesis. Moreover, combining chemotherapeutic and anti-senescence agents (or senolytic drugs) allows the selective elimination of senescent cells to restore tissue function, and potentially improve organ regeneration [7,141]. However, a major risk is that a population of post-chemotherapy senescent cancer cells may at some point re-enter the cell cycle, start to proliferate, become highly aggressive and chemo-resistant, and acquire stemness-like characteristics, leading to cancer relapse [366,368]. As many anti-cancer agents trigger cellular senescence in tumors, it is important to uncover the molecular mechanisms involved in senescence escape in order to propose novel therapies for controlling cellular senescence and maximizing therapeutic benefit and favorable outcomes.

## Figures and Tables

**Figure 2 ijms-22-13173-f002:**
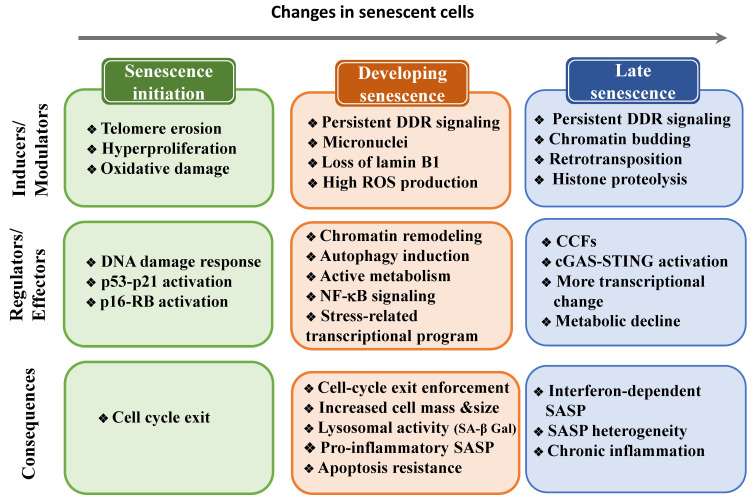
Summary of the most common senescence inducers and alterations observed at initiation, developing, and late senescence. CCFs, cytoplasmic chromatin fragments; cGAS-STING, cyclic GMP–AMP synthase-stimulator of interferon genes; DDR, DNA damage response; ROS, reactive oxygen species; SA-βGal, senescence-associated β galactosidase; SASP, senescence-associated secretory phenotype.

**Figure 3 ijms-22-13173-f003:**
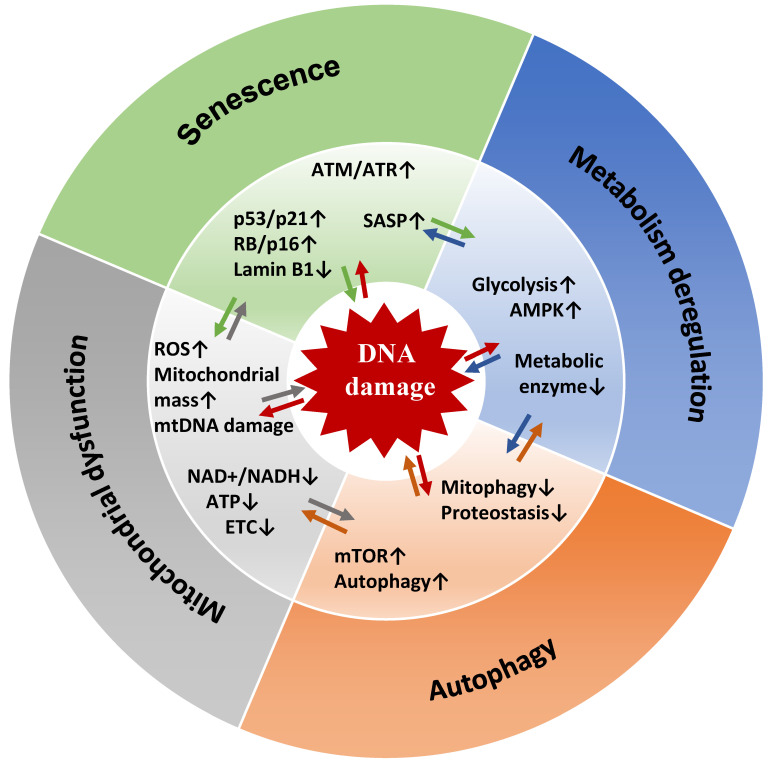
Overview of the mechanisms by which DNA damage promotes senescence. DNA damage (through telomere dysfunction and replicative stress) can result in cellular senescence, mitochondrial dysfunction, autophagy defects, and metabolic changes. These functional alterations are all interconnected and generate positive feedback signals that induce more DNA damage. This creates a cycle that contributes to stabilize the senescence-related cell cycle exit. AMPK, adenosine monophosphate-activated protein kinase; ETC, electron transport chain; mtDNA, mitochondrial DNA; mTOR, mammalian target of rapamycin; ROS, reactive oxygen species; SASP, senescence-associated secretory phenotype.

**Figure 4 ijms-22-13173-f004:**
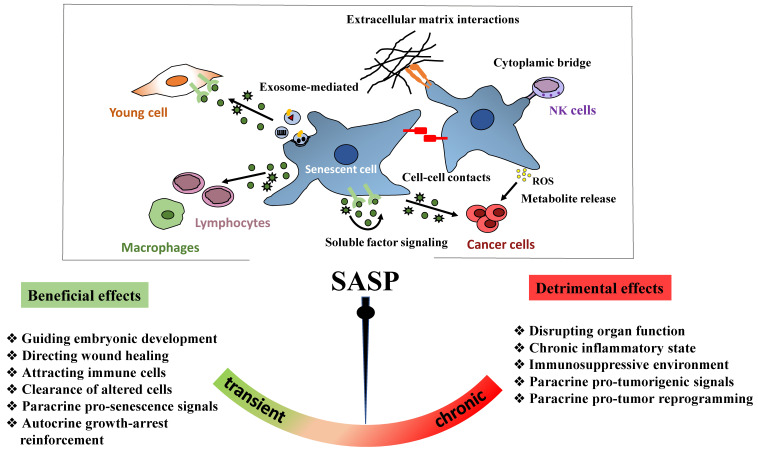
SASP signaling controls the beneficial and detrimental functions of senescent cells. Senescent cells through their secretome are actively engaged in cell-to-cell communications and extracellular matrix remodeling within the tissue microenvironment. The scheme depicts how senescent cells communicate *via* soluble factors, the release of extracellular vesicles (exosomes), cell–cell contacts, formation of cytoplasmic bridges, interactions with the extracellular matrix, and secretion of metabolites. It also lists some of the SASP-associated functions. A transient SASP is beneficial, while a chronic SASP causes negative outcomes. NK cells, natural killer cells; SASP, senescence-associated secretory phenotype.

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
