# Peer review of "Mechanisms and Regulation of Cellular Senescence"

_ijms, 2021, doi:10.3390/ijms222313173_

Round 1
Reviewer 1 Report
This is a very wordy review of many aspects of cell senescence. The review of the literature was extensive and thorough. Some of the paragraphs within the sections are overly long making the information harder to read and evaluate. The authors did not avoid repetitions and redundancy. This especially concerns the information about irreversible/permanent proliferation arrest, the role of telomeres and DDR in cellular senescence and the role of cell senescence as a cancer barrier. I strongly suggest to re-read carefully the manuscript and skip repetitions. I would be more careful writing that senescent cells are characterized by permanent or/ and irreversible proliferation arrest. Especially in the case of senescent cancer cells. Although the authors mentioned about escaping senescence by cancer cells, but they completely ignored the growing body of evidence that senescent cells, not only cancer but also normal ones, can undergo polyploidization and come back to the normal cycling (e.g. 10.1016/j.semcancer.2021.10.006, 10.1016/j.semcancer.2020.12.010, 10.1016/j.arr.2021.101458, 10.1186/1475-2867-13-92).
Chapter 7. Dysregulation of Cellular Senescence is completely incomprehensible to me. What do the authors mean? That detrimental role of non-cleared by the immune system of senescent cells is a sort of dysregulation? The statement: “Indeed, when they are not efficiently cleared and accumulate in tissues (for instance in the presence of continuous damage), these non-functional cells that secrete uncontrolled amounts of SASP factors can contribute to the local inflammation and aggravate tissue dysfunction. Thus, if this well-orchestrated response is deregulated, the restoration of tissue homeostasis is impaired” is illogical. Moreover, I am not sure if SASP is out of control. SASP can change in time, can be different in various senescent cells and is strictly controlled by different mechanisms described in this manuscript.
Author Response
This is a very wordy review of many aspects of cell senescence. The review of the literature was extensive and thorough. Some of the paragraphs within the sections are overly long making the information harder to read and evaluate. The authors did not avoid repetitions and redundancy. This especially concerns the information about irreversible/permanent proliferation arrest, the role of telomeres and DDR in cellular senescence and the role of cell senescence as a cancer barrier. I strongly suggest to re-read carefully the manuscript and skip repetitions. I would be more careful writing that senescent cells are characterized by permanent or/ and irreversible proliferation arrest. Especially in the case of senescent cancer cells. Although the authors mentioned about escaping senescence by cancer cells, but they completely ignored the growing body of evidence that senescent cells, not only cancer but also normal ones, can undergo polyploidization and come back to the normal cycling (e.g. 10.1016/j.semcancer.2021.10.006, 10.1016/j.semcancer.2020.12.010, 10.1016/j.arr.2021.101458, 10.1186/1475-2867-13-92).
Reply: We carefully read the manuscript again and eliminated redundancies wherever possible. Of note, we also removed several paragraphs in order to better focus on the key points.
It is correct that in cancer cells the senescence-associated cell cycle arrest may not be irreversible as the major pathways responsible for the maintenance of the arrest are altered. Considering that the senescence markers used are neither very sensitive nor specific can we really say that cell cycle arrest in cancer cells is stable in the first instance? The reversibility of the cell cycle arrest might also occur in some normal cells arrested following ras-induced senescence. So, as pointed out, we have amended the parts of the text that said the senescence arrest is irreversible.
Chapter 7. Dysregulation of Cellular Senescence is completely incomprehensible to me. What do the authors mean? That detrimental role of non-cleared by the immune system of senescent cells is a sort of dysregulation? The statement: “Indeed, when they are not efficiently cleared and accumulate in tissues (for instance in the presence of continuous damage), these non-functional cells that secrete uncontrolled amounts of SASP factors can contribute to the local inflammation and aggravate tissue dysfunction. Thus, if this well-orchestrated response is deregulated, the restoration of tissue homeostasis is impaired” is illogical. Moreover, I am not sure if SASP is out of control. SASP can change in time, can be different in various senescent cells and is strictly controlled by different mechanisms described in this manuscript.
Reply: The criticisms raised are valid, the word “dysregulation” is not appropriate. The title of this section is now “Relevance of senescence in cancer development and aging”. We have also rewritten the statement mentioned above, and trust these changes will make the text clearer.
Reviewer 2 Report
The paper describes the Mechanisms and Regulation of Cellular Senescence in the cells. This is an important topic, however, here are some of the recommendations to improve the paper quality
- Explain or give subtopic to the state or conditions that induce the cells senescence
- Justify the regulatory signals or how senescent cells affect/ how to transmit the signals to new cells to inhibit the growth
- Try to add some points of senescent cells role in some acute diseases or any functional characters in the healing processes
Author Response
The paper describes the Mechanisms and Regulation of Cellular Senescence in the cells. This is an important topic, however, here are some of the recommendations to improve the paper quality
- Explain or give subtopic to the state or conditions that induce the cells senescence
- Justify the regulatory signals or how senescent cells affect/ how to transmit the signals to new cells to inhibit the growth
- Try to add some points of senescent cells role in some acute diseases or any functional characters in the healing processes
Reply to point 1 :
Thank you for this comment. We have now added in paragraph 3.1 a summary of the different classes of stress signals that can lead to senescence. We have also added several subtopics to improve the reading.
Reply to point 2 :
In paragraph 7.1.2, we have added additional information on the paracrine effect of senescence by mentioning the potential pathways responsible for this effect: “ Senescent cells would achieve this effect through locally secreted SASP factors (Acosta et al. 2013). Secreted IL-1 and TGF-β induce paracrine senescence through a mechanism that generates ROS and DDR signalling in adjacent cells (Hubackova et al. 2012; Acosta et al. 2013). The underlying mechanism of paracrine signaling remains to be investigated”.
Reply to point 3 :
In this review, we have chosen to preferentially discuss the significance of cellular senescence in aging and age-associated diseases. Therefore, in response to this point, we have now added a brief discussion of the potential role of senescent cells accumulation in COVID-19 to paragraph 7.1.2.